# RANKINGMATCH: DELVING INTO SEMI-SUPERVISED LEARNING WITH CONSISTENCY REGULARIZATION AND RANKING LOSS

## ABSTRACT

Semi-supervised learning (SSL) has played an important role in leveraging unlabeled data when labeled data is limited. One of the most successful SSL approaches is based on consistency regularization, which encourages the model to produce unchanged with perturbed input. However, there has been less attention spent on inputs that have the same label. Motivated by the observation that the inputs having the same label should have the similar model outputs, we propose a novel method, RankingMatch, that considers not only the perturbed inputs but also the similarity among the inputs having the same label. We especially introduce a new objective function, dubbed BatchMean Triplet loss, which has the advantage of computational efficiency while taking into account all input samples. Our RankingMatch achieves state-of-the-art performance across many standard SSL benchmarks with a variety of labeled data amounts, including 95.13% accuracy on CIFAR-10 with 250 labels, 77.65% accuracy on CIFAR-100 with 10000 labels, 97.76% accuracy on SVHN with 250 labels, and 97.77% accuracy on SVHN with 1000 labels. We also perform an ablation study to prove the efficacy of the proposed BatchMean Triplet loss against existing versions of Triplet loss.

## 1 INTRODUCTION

Supervised learning and deep neural networks have proved their efficacy when achieving outstanding successes in a wide range of machine learning domains such as image recognition, language modeling, speech recognition, or machine translation. There is an empirical observation that better performance could be obtained if the model is trained on larger datasets with more labeled data (Hestness et al., 2017; Mahajan et al., 2018; Kolesnikov et al., 2019; Xie et al., 2020; Raffel et al., 2019). However, data labeling is costly and human-labor-demanding, even requiring the participation of experts (for example, in medical applications, data labeling must be done by doctors).

In many real-world problems, it is often very difficult to create a large amount of labeled training data. Therefore, numerous studies have focused on how to leverage unlabeled data, leading to a variety of research fields like self-supervised learning (Doersch et al., 2015; Noroozi & Favaro, 2016; Gidaris et al., 2018), semi-supervised learning (Berthelot et al., 2019b; Nair et al., 2019; Berthelot et al., 2019a; Sohn et al., 2020), or metric learning (Hermans et al., 2017; Zhang et al., 2019). In self-supervised learning, pretext tasks are designed so that the model can learn meaningful information from a large number of unlabeled images. The model is then fine-tuned on a smaller set of labeled data. In another way, semi-supervised learning (SSL) aims to leverage both labeled and unlabeled data in a single training process. On the other hand, metric learning does not directly predict semantic labels of given inputs but aims to measure the similarity among inputs.

In this paper, we unify the idea of semi-supervised learning (SSL) and metric learning to propose RankingMatch, a more powerful SSL method for image classification (Figure 1). We adopt FixMatch SSL method (Sohn et al., 2020), which utilized pseudo-labeling and consistency regularization to produce artificial labels for unlabeled data. Specifically, given an unlabeled image, its weakly-augmented and strongly-augmented version are created. The model prediction corresponding to the weakly-augmented image is used as the target label for the strongly-augmented image, encouraging the model to produce the same prediction for different perturbations of the same input.

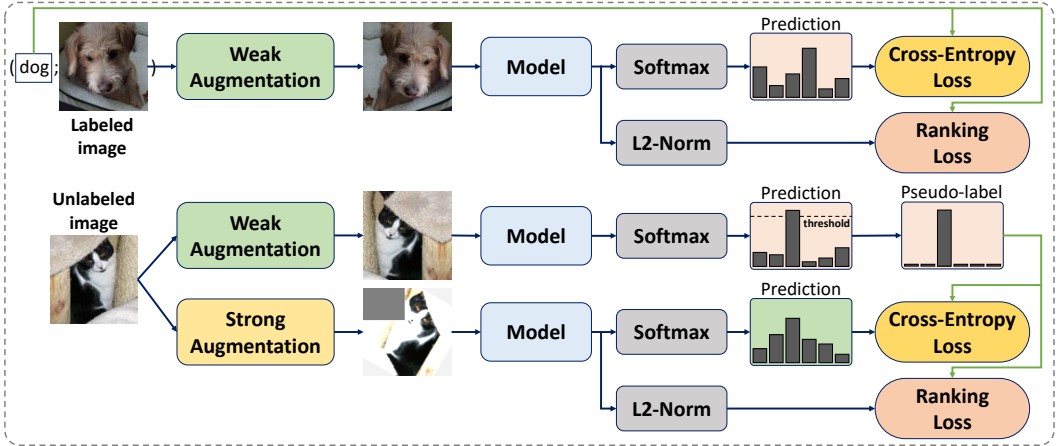

Figure 1: Diagram of RankingMatch. In addition to Cross-Entropy loss, Ranking loss is used to encourage the model to produce the similar outputs for the images from the same class.

Consistency regularization approach incites the model to produce unchanged with the different perturbations of the same input, but this is not enough. Inspired by the observation that the images from the same class (having the same label) should also have the similar model outputs, we utilize loss functions of metric learning, called Ranking losses, to apply more constraints to the objective function of our model. Concretely, we use Triplet and Contrastive loss with the aim of encouraging the model to produce the similar outputs for the images from the same class. Given an image from a class (saying *dog*, for example), Triplet loss tries to pull positive samples (images from class *dog*) nearer the given image and push negative samples (images not from class *dog*) further the given image. On the other hand, Contrastive loss maximizes the similarity of the images from the same class and minimizes the similarity of the images from different classes. However, instead of applying Triplet and Contrastive loss to the image representation as previous works did (Hermans et al., 2017; Chen et al., 2020a), we directly apply them to the model output (the "logits" score) which is the output of the classification head. *We argue that the images from the same class do not have to have similar representations strictly, but their model outputs should be as similar as possible.* Our motivation and argument could be consolidated in Appendix A. Especially, we propose a new version of Triplet loss which is called BatchMean. Our BatchMean Triplet loss has the advantage of computational efficiency of existing BatchHard Triplet loss while taking into account all input samples when computing the loss. More details will be presented in Section 3.3.1. Our key contributions are summarized as follows:

- We introduce a novel SSL method, RankingMatch, that encourages the model to produce the similar outputs for not only the different perturbations of the same input but also the input samples from the same class.
- Our proposed BatchMean Triplet loss surpasses two existing versions of Triplet loss which are BatchAll and BatchHard Triplet loss (Section 4.5).
- Our method is simple yet effective, achieving state-of-the-art results across many standard SSL benchmarks with various labeled data amounts.

## 2    RELATED WORK

Many recent works have achieved success in semi-supervised learning (SSL) by adding a loss term for unlabeled data. This section reviews two classes of this loss term (consistency regularization and entropy minimization) that are related to our work. Ranking loss is also reviewed in this section.

**Consistency Regularization**    This is a widely used SSL technique which encourages the model to produce unchanged with different perturbations of the same input sample. Consistency regularization was early introduced by Sajjadi et al. (2016) and Laine & Aila (2016) with the methods named "Regularization With Stochastic Transformations and Perturbations" and "Π-Model", respectively.

Both of these two approaches used Mean Squared Error (MSE) to enforce the model to produce the same output for different perturbed versions of the same input. Later state-of-the-art methods adopted consistency regularization in diverse ways. In MixMatch (Berthelot et al., 2019b), a guessed label, computed based on $K$ weakly-augmented versions of an unlabeled sample, was used as the target label for all these $K$ weakly-augmented samples. On the other hand, in FixMatch (Sohn et al., 2020), a pseudo-label, which is computed based on the weakly-augmented unlabeled sample, became the target label for the strongly-augmented version of the same unlabeled sample.

**Entropy Minimization**   One of the requirements in SSL is that the model prediction for unlabeled data should have low entropy. Grandvalet & Bengio (2005) and Miyato et al. (2018) introduced an additional loss term, which is explicitly incorporated in the objective function, to minimize the entropy of the distribution of the model prediction for unlabeled data. On the other hand, MixMatch (Berthelot et al., 2019b) used a sharpening function to adjust the model prediction distribution and thereby reduced the entropy of the predicted label. FixMatch (Sohn et al., 2020) implicitly obtained entropy minimization by constructing hard labels from high-confidence predictions (predictions which are higher than a pre-defined threshold) on weakly-augmented unlabeled data. These hard labels were then used as the target labels for strongly-augmented unlabeled data.

**Metric Learning and Ranking Loss**   Metric learning is an approach that does not directly predict semantic labels of given images but trains the model to learn the similarity among samples (Kulis et al., 2012; Kaya & Bilge, 2019). There are various objective functions used in metric learning, including Triplet and Contrastive loss which are used in our work. Triplet loss was successfully exploited in person re-identification problem (Hermans et al., 2017). A triplet contains a person image referred to as anchor, a positive sample which is the image from the same person with the anchor, and a negative sample being the image from the different person with the anchor. Triplet loss was used to enforce the distance between the anchor and negative sample to be larger than the distance between the anchor and positive sample by at least a margin $m$. Besides, SimCLR (Chen et al., 2020a) utilized Contrastive loss to maximize the similarity between two different augmented versions of the same sample while minimizing the similarity between different samples. Both Hermans et al. (2017) and Chen et al. (2020a) applied Triplet and Contrastive loss to the image representation. Contrastive loss was also used by Chen et al. (2020b) for semi-supervised image retrieval and person re-identification. Given feature (or image) representations, Chen et al. (2020b) computed class-wise similarity scores using a similarity measurement to learn semantics-oriented similarity representation. Contrastive loss was then applied to both image and semantics-oriented similarity representation in two learning phases. If the model output in image classification is viewed as a form of class-wise similarity scores, the high-level idea of our method might be similar to Chen et al. (2020b) in utilizing Contrastive loss. However, in our case, the model itself obtains class-wise similarity scores, and Contrastive loss is only applied to the model output ("logits" score, but not image representation) in a single training process. More details will be presented in Section 3.3.

# 3   RANKINGMATCH

This section starts to describe the overall framework and objective function of RankingMatch. Next, two important factors of the objective function, Cross-Entropy and Ranking loss, will be presented in detail. Concretely, Triplet and Contrastive loss will be separately shown with our proposed and modified versions.

## 3.1   OVERALL FRAMEWORK

The overall framework of RankingMatch is illustrated in Figure 1. Both labeled and unlabeled data are simultaneously leveraged in a single training process. Two kinds of augmentation are used to perturb the input sample. While weak augmentation uses standard padding-and-cropping and horizontal flipping augmentation strategies, more complex transformations are used for strong augmentation. We utilize RandAugment (Cubuk et al., 2020) for strong augmentation, consisting of multiple transformation methods such as contrast adjustment, shear, rotation, translation, etc. Given a collection of transformations, two of them are randomly selected to strongly perturb the input sample. Cutout (DeVries & Taylor, 2017) is followed to obtain the final strongly-augmented sample.

As shown in Figure 1, only weak augmentation is used for labeled data. The weakly-augmented labeled image is fed into the model to produce scores for labels. These scores are actually the output of the classification head, and we call them "logits" score for a convenient explanation. A softmax function is used to convert the "logits" scores to the probabilities for labels. These probabilities are then used along with ground-truth labels to compute Cross-Entropy loss. An $L_2$-normalization is applied to the "logits" scores before using them for computing Ranking loss. We experimented and found that $L_2$-normalization is an important factor contributing to the success of RankingMatch, which will be shown in Section 4.5. The ground-truth labels are used to determine positive samples (images from the same class) and negative samples (images from different classes) in computing Ranking loss. The same procedure is used for unlabeled data except that pseudo-labels, obtained from weakly-augmented unlabeled samples, are used instead of the ground-truth labels.

Let $\mathcal{X} = \{(x_b, l_b) : b \in (1, ..., B)\}$ define a batch of $B$ labeled samples, where $x_b$ is the training sample and $l_b$ is the corresponding one-hot label. Let $\mathcal{U} = \{u_b : b \in (1, ..., \mu B)\}$ be a batch of $\mu B$ unlabeled samples with a coefficient $\mu$ determining the relative size of $\mathcal{X}$ and $\mathcal{U}$. We denote weak and strong augmentation as $\mathcal{A}_w(.)$ and $\mathcal{A}_s(.)$ respectively. Let $p_{\text{model}}(y \mid x; \theta)$ be the "logits" score produced by the model for a given input $x$. As a result, $\text{Softmax}(p_{\text{model}}(y \mid x; \theta))$ and $\text{L2Norm}(p_{\text{model}}(y \mid x; \theta))$ are the softmax function and $L_2$-normalization applied to the "logits" score, respectively. Finally, let $\text{H}(v, q)$ be Cross-Entropy loss of the predicted class distribution $q$ and the target label $v$. Notably, $v$ corresponds to the ground-truth label or pseudo-label in the case of labeled or unlabeled data respectively.

As illustrated in Figure 1, there are four elements contributing to the overall loss function of RankingMatch. Two of them are Cross-Entropy loss for labeled and unlabeled data, denoted by $\mathcal{L}_s^{CE}$ and $\mathcal{L}_u^{CE}$ respectively. Two remaining ones are Ranking loss for labeled and unlabeled data, corresponding to $\mathcal{L}_s^{Rank}$ and $\mathcal{L}_u^{Rank}$ respectively. The objective is minimizing the loss function defined as follows:

$$\mathcal{L} = \mathcal{L}_s^{CE} + \lambda_u \mathcal{L}_u^{CE} + \lambda_r(\mathcal{L}_s^{Rank} + \mathcal{L}_u^{Rank}) \tag{1}$$

where $\lambda_u$ and $\lambda_r$ are scalar hyperparameters denoting the weights of the loss elements. In Section 3.2 and 3.3, we will present how Cross-Entropy and Ranking loss are computed for labeled and unlabeled data in detail. We also show comparisons between RankingMatch and other methods in Appendix B. The full algorithm of RankingMatch is provided in Appendix C.

## 3.2 CROSS-ENTROPY LOSS

For labeled data, since the ground-truth labels are available, the standard Cross-Entropy loss is computed as follows:

$$\mathcal{L}_s^{CE} = \frac{1}{B} \sum_{b=1}^{B} \text{H}(l_b, \text{Softmax}(p_{\text{model}}(y \mid \mathcal{A}_w(x_b); \theta))) \tag{2}$$

For unlabeled data, we adopt the idea of FixMatch (Sohn et al., 2020) to obtain the pseudo-label which plays the similar role as the ground-truth label of labeled data. Given an unlabeled image $u_b$, the model first produces the "logits" score for the weakly-augmented unlabeled image: $q_b = p_{\text{model}}(y \mid \mathcal{A}_w(u_b); \theta)$. A softmax function is then applied to $q_b$ to obtain the model prediction: $\tilde{q}_b = \text{Softmax}(q_b)$. The pseudo-label corresponds to the class having the highest probability: $\hat{q}_b = \text{argmax}(\tilde{q}_b)$. Note that for simplicity, $\text{argmax}$ is assumed to produce the valid one-hot pseudo-label. A threshold $\tau$ is used to ignore predictions that have low confidence. Finally, the high-confidence pseudo-labels are used as the target labels for strongly-augmented versions of corresponding unlabeled images, leading to:

$$\mathcal{L}_u^{CE} = \frac{1}{\mu B} \sum_{b=1}^{\mu B} \mathbb{1}(\max(\tilde{q}_b) \geq \tau) \, \text{H}(\hat{q}_b, \text{Softmax}(p_{\text{model}}(y \mid \mathcal{A}_s(u_b); \theta))) \tag{3}$$

Equation 3 satisfies consistency regularization and entropy minimization. The model is encouraged to produce consistent outputs for strongly-augmented samples against the model outputs for weakly-augmented samples; this is referred to as consistency regularization. As advocated in Lee (2013) and Sohn et al. (2020), the use of a pseudo-label, which is based on the model prediction for an unlabeled sample, as a hard target for the same sample could be referred to as entropy minimization.

### 3.3 RANKING LOSS

This section presents two types of Ranking loss used in our RankingMatch, which are Triplet and Contrastive loss. We directly apply these two loss functions to the "logits" scores, which is different from previous works such as Hermans et al. (2017) and Chen et al. (2020a). Especially, our novel version of Triplet loss, which is BatchMean Triplet loss, will also be presented in this section.

Let $\mathcal{C}$ be a batch of $L_2$-normalized "logits" scores of the network shown in Figure 1. Let $y_i$ denote the label of the $L_2$-normalized "logits" score $i$. This label could be the ground-truth label or pseudo-label in the case of labeled or unlabeled data, respectively. The procedure of obtaining the pseudo-label for unlabeled data was presented in Section 3.2. Notably, Ranking loss is separately computed for labeled and unlabeled data, $\mathcal{L}_s^{Rank}$ and $\mathcal{L}_u^{Rank}$ in Equation 1 could be either Triplet loss (Section 3.3.1) or Contrastive loss (Section 3.3.2). Let $a$, $p$, and $n$ be the anchor, positive, and negative sample, respectively. While the anchor and positive sample represent the $L_2$-normalized "logits" scores having the same label, the anchor and negative sample are for the $L_2$-normalized "logits" scores having the different labels.

#### 3.3.1 BATCHMEAN TRIPLET LOSS

Let $d_{i,j}$ denote the distance between two "logits" scores $i$ and $j$. Following Schroff et al. (2015) and Hermans et al. (2017), two existing versions of Triplet loss, BatchAll and BatchHard, could be defined as follows with the use of Euclidean distance for $d_{i,j}$.

BatchAll Triplet loss:

$$\mathcal{L}_{\text{BA}} = \frac{1}{V} \sum_{\substack{a,p,n\in\mathcal{C} \\ y_a=y_p\neq y_n}} f(m + d_{a,p} - d_{a,n}) \tag{4}$$

where $V$ is the number of triplets. A triplet consists of an anchor, a positive sample, and a negative sample.

BatchHard Triplet loss:

$$\mathcal{L}_{\text{BH}} = \frac{1}{|\mathcal{C}|} \sum_{a\in\mathcal{C}} f(m + \max_{\substack{p\in\mathcal{C} \\ y_p=y_a}} d_{a,p} - \min_{\substack{n\in\mathcal{C} \\ y_n\neq y_a}} d_{a,n}) \tag{5}$$

In Equation 4 and 5, $m$ is the margin, and $f(\bullet)$ indicates the function to avoid revising "already correct" triplets. A hinge function ($f(\bullet) = \max(0, \bullet)$) could be used in this circumstance. For instance, if a triplet already satisfied the distance between the anchor and negative sample is larger than the distance between the anchor and positive sample by at least a margin $m$, that triplet should be ignored from the training process by assigning it zero-value ($f(m+d_{a,p}-d_{a,n}) = 0$ if $m+d_{a,p}-d_{a,n} \leq 0$, corresponding to $d_{a,n} - d_{a,p} \geq m$). However, as mentioned in Hermans et al. (2017), the softplus function ($\ln(1 + \exp(\bullet))$) gives better results compared to the hinge function. Thus, we decided to use the softplus function for all our experiments, which is referred to as soft-margin.

While BatchAll considers all triplets, BatchHard only takes into account hardest triplets. A hardest triplet consists of an anchor, a furthest positive sample, and a nearest negative sample relative to that anchor. The intuition behind BatchHard is that if we pull an anchor and its furthest positive sample together, other positive samples of that anchor will also be pulled obviously. BatchHard is more computationally efficient compared to BatchAll. However, because max and min function are used in BatchHard, only the hardest triplets (anchors, furthest positive samples, and nearest negative samples) are taken into account when the network does backpropagation. We argue that it would be beneficial if all samples are considered and contribute to updating the network parameters. Therefore, we introduce a novel variant of Triplet loss, called **BatchMean** Triplet loss, as follows:

$$\mathcal{L}_{\text{BM}} = \frac{1}{|\mathcal{C}|} \sum_{a\in\mathcal{C}} f(m + \frac{1}{|\mathcal{C}|} \sum_{\substack{p\in\mathcal{C} \\ y_p=y_a}} d_{a,p} - \frac{1}{|\mathcal{C}|} \sum_{\substack{n\in\mathcal{C} \\ y_n\neq y_a}} d_{a,n}) \tag{6}$$

By using "mean" function ($\frac{1}{|\mathcal{C}|}\sum_{\mathcal{C}}$) instead of max and min function, our proposed BatchMean Triplet loss not only has the advantage of computational efficiency of BatchHard but also takes into account all samples. The efficacy of BatchMean Triplet loss will be clarified in Section 4.5.

### 3.3.2 CONTRASTIVE LOSS

Let $sim_{i,j}$ denote the similarity between two $L_2$-normalized "logits" scores $i$ and $j$. Referring to Chen et al. (2020a), we define Contrastive loss applied to our work as follows:

$$\mathcal{L}_{CT} = \frac{1}{N} \sum_{\substack{a,p \in \mathcal{C} \\ a \neq p, y_a = y_p}} -\ln \frac{\exp\left(sim_{a,p}/T\right)}{\exp\left(sim_{a,p}/T\right) + \sum_{\substack{n \in \mathcal{C} \\ y_n \neq y_a}} \exp\left(sim_{a,n}/T\right)} \tag{7}$$

where $N$ is the number of valid pairs of anchor and positive sample, and $T$ is a constant denoting the temperature parameter. Note that if the $i^{th}$ and $j^{th}$ "logits" score of $\mathcal{C}$ have the same label, there will be two valid pairs of anchor and positive sample. The $i^{th}$ "logits" score could become an anchor, and the $j^{th}$ "logits" score is a positive sample; and vice versa. The form of $\mathcal{L}_{CT}$ is referred to as the normalized temperature-scaled cross-entropy loss. The objective is minimizing $\mathcal{L}_{CT}$; this corresponds to maximizing $sim_{a,p}$ and minimizing $sim_{a,n}$. Moreover, we also want the anchor and positive sample to be as similar as possible. As a result, cosine similarity is a suitable choice for the similarity function of $\mathcal{L}_{CT}$. For instance, if two "logits" score vectors are the same, the cosine similarity between them has the maximum value which is $1$.

## 4 EXPERIMENTS

We evaluate the efficacy of RankingMatch on standard semi-supervised learning (SSL) benchmarks such as CIFAR-10 (Krizhevsky et al., 2009), CIFAR-100 (Krizhevsky et al., 2009), SVHN (Netzer et al., 2011), and STL-10 (Coates et al., 2011). We also conduct experiments on Tiny ImageNet[1] to verify the performance of our method on a larger dataset. Our method is compared against Mix-Match (Berthelot et al., 2019b), RealMix (Nair et al., 2019), ReMixMatch (Berthelot et al., 2019a), and FixMatch (Sohn et al., 2020). As recommended by Oliver et al. (2018), all methods should be implemented using the same codebase. However, due to the limited computing resources, we only re-implemented MixMatch and FixMatch. Our target is not reproducing state-of-the-art results of these papers, but making the comparison with our method as fair as possible.

### 4.1 IMPLEMENTATION DETAILS

Unless otherwise noted, we utilize Wide ResNet-28-2 network architecture (Zagoruyko & Komodakis, 2016) with $1.5$ million parameters, and our experiments are trained for 128 epochs with a batch size of 64. Concretely, for our RankingMatch, we use a same set of hyperparameters ($B = 64$, $\mu = 7$, $\tau = 0.95$, $m = 0.5$, $T = 0.2$, $\lambda_u = 1$, and $\lambda_r = 1$) across all datasets and all amounts of labeled samples except that a batch size of 32 ($B = 32$) is used for the STL-10 dataset. More details of the training protocol and hyperparameters will be reported in Appendix D. In all our experiments, FixMatch[RA] and FixMatch[CTA] refer to FixMatch with using RandAugment and CTAugment respectively (Sohn et al., 2020); RankingMatch[BM], RankingMatch[BH], RankingMatch[BA], and RankingMatch[CT] refer to RankingMatch with using BatchMean Triplet loss, BatchHard Triplet loss, BatchAll Triplet loss, and Contrastive loss respectively. For each benchmark dataset, our results are reported on the corresponding test set.

### 4.2 CIFAR-10 AND CIFAR-100

**Results with same settings** We first implement all methods using the same codebase and evaluate them under same conditions to show how effective our method could be. The results are shown in Table 1. Note that different folds mean different random seeds. As shown in Table 1, RankingMatch outperforms all other methods across all numbers of labeled samples, especially with a small portion of labels. For example, on CIFAR-10, RankingMatch[BM] with 40 labels reduces the error rate by 29.61% and 4.20% compared to MixMatch and FixMatch[RA] respectively. The results also show that cosine similarity might be more suitable than Euclidean distance if the dimension of the "logits" score grows up. For instance, on CIFAR-100 where the "logits" score is a 100-dimensional vector, RankingMatch[CT] reduces the error rate by 1.07% and 1.19% compared to RankingMatch[BM] in the case of 2500 and 10000 labels respectively.

---

[1]Stanford University. http://cs231n.stanford.edu/

Table 1: Error rates (%) for CIFAR-10 and CIFAR-100 on five different folds. All methods are implemented using the same codebase.

| Method | CIFAR-10 | | | CIFAR-100 | | |
|---|---|---|---|---|---|---|
| | 40 labels | 250 labels | 4000 labels | 400 labels | 2500 labels | 10000 labels |
| MixMatch | 44.83±8.70 | 19.46±1.25 | 7.74±0.21 | 82.10±0.78 | 48.98±0.88 | 35.11±0.36 |
| FixMatch[RA] | 19.42±6.46 | 7.30±0.79 | 4.84±0.23 | 61.02±1.61 | 38.17±0.40 | 30.23±0.43 |
| RankingMatch[BM] | **15.22**±4.51 | **6.77**±0.89 | **4.76**±0.17 | **60.59**±2.05 | 38.26±0.39 | 30.46±0.24 |
| RankingMatch[CT] | **16.66**±2.77 | **7.26**±1.20 | **4.81**±0.33 | 64.26±0.80 | **37.19**±0.55 | **29.27**±0.30 |

**CIFAR-10 with more training epochs** Since FixMatch, which our RankingMatch is based on, was trained for 1024 epochs, we attempted to train our models with more epochs to make our results comparable. Our results on CIFAR-10, which were trained for 256 epochs, are reported on the left of Table 2. RankingMatch[BM] achieves a state-of-the-art result, which is 4.87% error rate with 250 labels, even trained for fewer epochs compared to FixMatch. With 40 labels, RankingMatch[BM] has worse performance compared to FixMatch[CTA] but reduces the error rate by 0.38% compared to FixMatch[RA]. Note that both RankingMatch[BM] and FixMatch[RA] use RandAugment, so the results are comparable.

**CIFAR-100 with a larger model** We use a larger version of Wide ResNet-28-2 network architecture, which is called Wide ResNet-28-2-Large, by using more filters per layer. Specifically, the number of filters of layers in the second group is increased from 32 to 135, and similarly for the following groups with a multiplication factor of 2. This Wide ResNet-28-2-Large network architecture with 26 million parameters was also used in MixMatch and FixMatch, so making our results comparable. As shown on the right of Table 2, RankingMatch[CT] achieves a state-of-the-art result, which is 22.35% error rate with 10000 labels, even trained for only 128 epochs.

Table 2: Comparison with state-of-the-art methods in error rate (%) on CIFAR-10 and CIFAR-100. Methods marked by * denote results cited from respective papers. On CIFAR-10, our models are trained for 256 epochs. For CIFAR-100, Wide ResNet-28-2-Large network architecture is used.

| Method | CIFAR-10 | | | CIFAR-100 | | |
|---|---|---|---|---|---|---|
| | 40 labels | 250 labels | 4000 labels | 400 labels | 2500 labels | 10000 labels |
| MixMatch* | - | 11.08±0.87 | 6.24±0.06 | - | - | 25.88±0.30 |
| RealMix* | - | 9.79±0.75 | 6.39±0.27 | - | - | - |
| ReMixMatch* | - | 6.27±0.34 | 5.14±0.04 | - | - | - |
| FixMatch[RA]* | 13.81±3.37 | 5.07±0.65 | **4.26**±0.05 | **48.85**±1.75 | **28.29**±0.11 | 22.60±0.12 |
| FixMatch[CTA]* | **11.39**±3.35 | 5.07±0.33 | 4.31±0.15 | 49.95±3.01 | 28.64±0.24 | 23.18±0.11 |
| RankingMatch[BM] | 13.43±2.33 | **4.87**±0.08 | 4.29±0.03 | 49.57±0.67 | 29.68±0.60 | 23.18±0.03 |
| RankingMatch[CT] | 14.98±3.06 | 5.13±0.02 | 4.32±0.12 | 56.90±1.47 | 28.39±0.67 | **22.35**±0.10 |

## 4.3 SVHN AND STL-10

**SVHN** The results for SVHN are shown in Table 3. We achieve state-of-the-art results, which are 2.24% and 2.23% error rate in the case of 250 and 1000 labels, respectively. With 40 labels, our results are worse than those of FixMatch; this may be excusable because our models were trained for 128 epochs while FixMatch's models were trained for 1024 epochs.

**STL-10** STL-10 is a dataset designed for unsupervised learning, containing 5000 labeled images and 100000 unlabeled images. To deal with the higher resolution of images in the STL-10 dataset ($96 \times 96$), we add one more group to the Wide ResNet-28-2 network, resulting in Wide ResNet-37-2 architecture with 5.9 million parameters. There are ten pre-defined folds with 1000 labeled images each. Table 4 shows our results on three of these ten folds. The result of SWWAE and CC-GAN are cited from Zhao et al. (2015) and Denton et al. (2016) respectively. We achieve better results

compared to numerous methods. Our RankingMatch$^{BM}$ obtains an error rate of 5.96% while the current state-of-the-art method (FixMatch) has the error rate of 7.98% and 5.17% in the case of using RandAugment and CTAugment respectively.

Table 3: Comparison with state-of-the-art methods in error rate (%) on SVHN. * denotes the results cited from respective papers. Our results are reported on five different folds.

| Method | 40 labels | 250 labels | 1000 labels |
|---|---|---|---|
| MixMatch* | - | 3.78±0.26 | 3.27±0.31 |
| RealMix* | - | 3.53±0.38 | - |
| ReMixMatch* | - | 3.10±0.50 | 2.83±0.30 |
| FixMatch$^{RA}$* | **3.96**±2.17 | 2.48±0.38 | 2.28±0.11 |
| FixMatch$^{CTA}$* | 7.65±7.65 | 2.64±0.64 | 2.36±0.19 |
| MixMatch | 42.55±15.94 | 6.25±0.17 | 5.87±0.12 |
| FixMatch$^{RA}$ | 24.95±10.29 | 2.37±0.26 | 2.28±0.12 |
| RankingMatch$^{BM}$ | 21.02±8.06 | **2.24**±0.07 | 2.32±0.07 |
| RankingMatch$^{CT}$ | 27.20±2.90 | 2.33±0.06 | **2.23**±0.11 |

Table 4: Error rates (%) for STL-10 on 1000-label splits. * denotes the results cited from respective papers.

| Method | Error Rate |
|---|---|
| SWWAE* | 25.67 |
| CC-GAN* | 22.21 |
| MixMatch* | 10.18±1.46 |
| ReMixMatch* | 6.18±1.24 |
| FixMatch$^{RA}$* | 7.98±1.50 |
| FixMatch$^{CTA}$* | **5.17**±0.63 |
| FixMatch$^{RA}$ | 6.10±0.11 |
| RankingMatch$^{BM}$ | **5.96**±0.07 |
| RankingMatch$^{CT}$ | 7.55±0.37 |

## 4.4 TINY IMAGENET

Tiny ImageNet is a compact version of ImageNet, consisting of 200 classes. We use the Wide ResNet-37-2 architecture with 5.9 million parameters, as used for STL-10. Only 9000 out of 100000 training images are used as labeled data. While FixMatch$^{RA}$ obtains an error rate of 52.09±0.14%, RankingMatch$^{BM}$ and RankingMatch$^{CT}$ achieve the error rate of 51.47±0.25% and 49.10±0.41% respectively (reducing the error rate by 0.62% and 2.99% compared to FixMatch$^{RA}$, respectively). Moreover, RankingMatch$^{CT}$ has the better result compared to RankingMatch$^{BM}$, advocating the efficiency of cosine similarity in a high-dimensional space, as presented in Section 4.2.

## 4.5 ABLATION STUDY

Table 5: Ablation study. All results are error rates (%). Results of the same label split (same column) are reported using the same seed. In the second row, NaN-70 means the loss becomes NaN after 70 training steps, and similarly for other cases. $^\dagger$ means the methods without $L_2$-normalization.

| | CIFAR-10 | | CIFAR-100 | | SVHN | |
|---|---|---|---|---|---|---|
| Ablation | 250 labels | 4000 labels | 2500 labels | 10000 labels | 250 labels | 1000 labels |
| RankingMatch$^{BM}$ | **5.50** | **4.49** | **37.79** | **30.14** | **2.11** | **2.23** |
| RankingMatch$^{BM\dagger}$ | NaN-70 | NaN-310 | NaN-730 | NaN-580 | NaN-30 | 2.29 |
| RankingMatch$^{BH}$ | 11.96 | 8.59 | 38.83 | 31.19 | 3.13 | 2.72 |
| RankingMatch$^{BA}$ | 24.17 | 12.05 | 49.06 | 34.96 | 3.00 | 3.45 |
| RankingMatch$^{CT}$ | **5.76** | **4.64** | **36.53** | **28.99** | **2.25** | **2.19** |
| RankingMatch$^{CT\dagger}$ | 6.31 | 4.67 | 36.77 | 29.15 | 2.43 | 2.20 |

We carried out the ablation study, as summarized in Table 5. Our BatchMean Triplet loss outperforms BatchHard and BatchAll Triplet loss. For example, on CIFAR-10 with 250 labels, RankingMatch$^{BM}$ reduces the error rate by a factor of two and four compared to RankingMatch$^{BH}$ and RankingMatch$^{BA}$, respectively. We also show the efficacy of $L_2$-normalization which contributes to the success of RankingMatch. Applying $L_2$-normalization helps reduce the error rate of RankingMatch$^{CT}$ across all numbers of labels. Especially for RankingMatch$^{BM}$, if $L_2$-normalization is not used, the model may not converge due to the very large value of the loss.

### 4.6 ANALYSIS

**Qualitative results**  We visualize the "logits" scores of the methods, as shown in Figure 2. While MixMatch has much more overlapping points, the other three methods have class *cyan* (*cat*) and *yellow* (*dog*) close together. Interestingly, RankingMatch has the shape of the clusters being different from MixMatch and FixMatch; this might open a research direction for future work. More details about our visualization are presented in Appendix E.

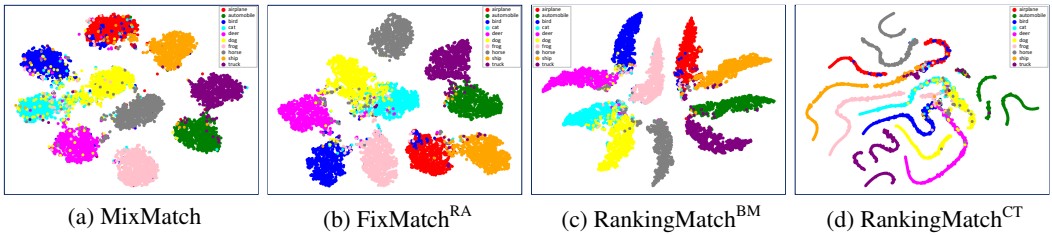

(a) MixMatch     (b) FixMatch$^{RA}$     (c) RankingMatch$^{BM}$     (d) RankingMatch$^{CT}$

Figure 2: t-SNE visualization of the "logits" scores of the methods on CIFAR-10 test set. The models were trained for 128 epochs with 4000 labels. The same color means the same class.

**Computational efficiency**  Figure 3 shows the training time per epoch of RankingMatch with using BatchAll, BatchHard, or BatchMean Triplet loss on CIFAR-10, SVHN, and CIFAR-100. While the training time of BatchHard and BatchMean is stable and unchanged among epochs, BatchAll has the training time gradually increased during the training process. Moreover, BatchAll requires more training time compared to BatchHard and BatchMean. For example, on SVHN, BatchAll has the average training time per epoch is 434.05 seconds, which is 126.25 and 125.82 seconds larger than BatchHard and BatchMean respectively.

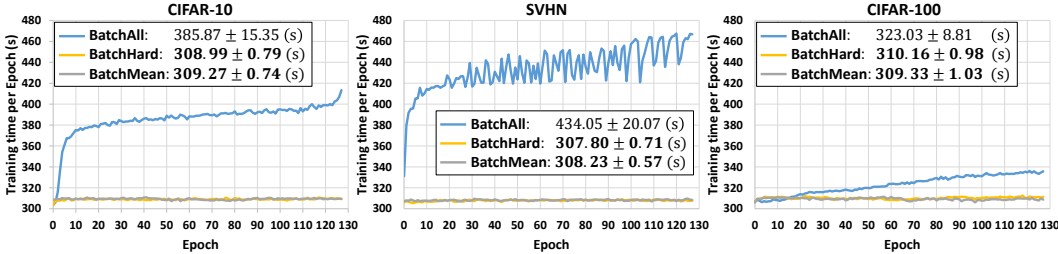

Figure 3: Training time per epoch (seconds) during 128 epochs.

We also measure the GPU memory usage of the methods during the training process. On average, BatchAll occupies two times more GPU memory than BatchHard and BatchMean. For instance, on CIFAR-10, the GPU memory usage of BatchAll is 9039.72±2043.30MB, while this value is 4845.92±0.72MB in BatchHard and BatchMean. More details are presented in Appendix F.

## 5 CONCLUSION

In this paper, we propose RankingMatch, a novel semi-supervised learning (SSL) method that unifies the idea of consistency regularization SSL approach and metric learning. Our method encourages the model to produce the same prediction for not only the different augmented versions of the same input but also the samples from the same class. Delving into the objective function of metric learning, we introduce a new variant of Triplet loss, called BatchMean Triplet loss, which has the advantage of computational efficiency while taking into account all samples. The extensive experiments show that our method exhibits good performance and achieves state-of-the-art results across many standard SSL benchmarks with various labeled data amounts. For future work, we are interested in researching the combination of Triplet and Contrastive loss in a single objective function so that we can take the advantages of these two loss functions.

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

## A    DETAILS OF OUR MOTIVATION AND ARGUMENT

**For our motivation of utilizing Ranking loss in semi-supervised image classification**    FixMatch (Sohn et al., 2020) is a simple combination of existing semi-supervised learning (SSL) approaches such as consistency regularization and pseudo-labeling. FixMatch, as well as the consistency regularization approach, only considers the different perturbations of the same input. The model should produce unchanged with the different perturbations of the same input, but this is not enough. Our work is to fulfill this shortcoming. Our main motivation is that the different inputs of the same class (for example, two different *cat* images) should also have the similar model outputs. We showed that by simply integrating Ranking loss (especially our proposed BatchMean Triplet loss) into FixMatch, we could achieve the promising results, as quantitatively shown in Section 4.

**For our argument**    We argue that *the images from the same class do not have to have similar representations strictly, but their model outputs should be as similar as possible*. Our work aims to solve the image classification task. Basically, the model for image classification consists of two main parts: feature extractor and classification head. Given an image, the feature extractor is responsible for understanding the image and generates the image representation. The image representation is then fed into the classification head to produce the model output (the "logits" score) which is the scores for all classes.

- If the feature extractor can generate the very similar image representations for the images from the same class, it will be beneficial for the classification head.
- Otherwise, if these image representations are not totally similar, the classification head will have to pay more effort to produce the similar model outputs for the same-class images.

Therefore, the model outputs somehow depend on the image representations. For image classification, *the goal is to get the similar model outputs for the same-class images even when the image representations are not totally similar*. That is also the main motivation for us to apply Ranking loss directly to the model outputs. Figure 4 illustrates the image representations and model outputs of the model when given same-class images.

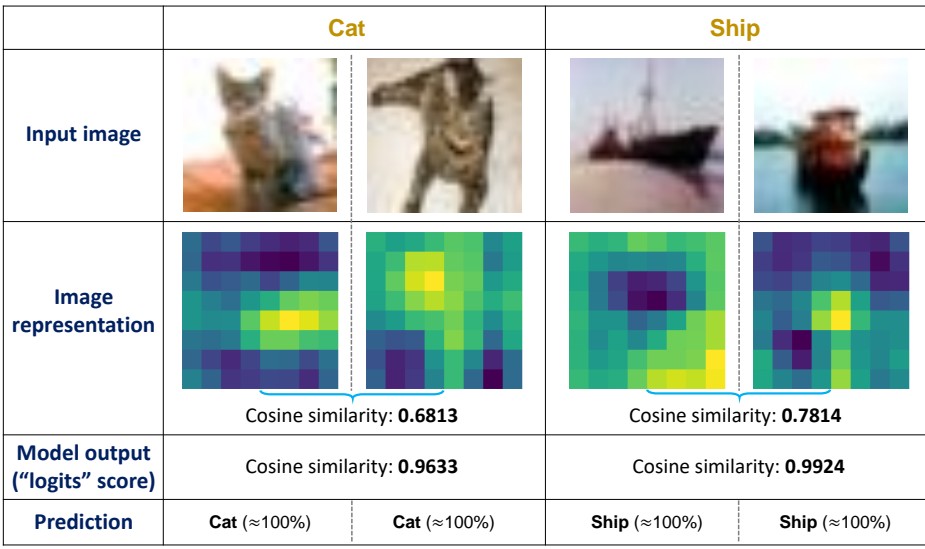

Figure 4: Illustration of image representation and model output on CIFAR-10.

As shown in Figure 4, given two images from the same class, although the model can exactly predict the semantic labels and get the very similar model outputs, the image representations are not totally similar. For instance, two *cat* images can have the model outputs with the cosine similarity of 0.9633, but the cosine similarity of two corresponding image representations is only 0.6813. To support why applying Ranking loss directly to the model outputs is beneficial, we visualize the image representations and model outputs of our method on the CIFAR-10 dataset, as shown in Figure 5.

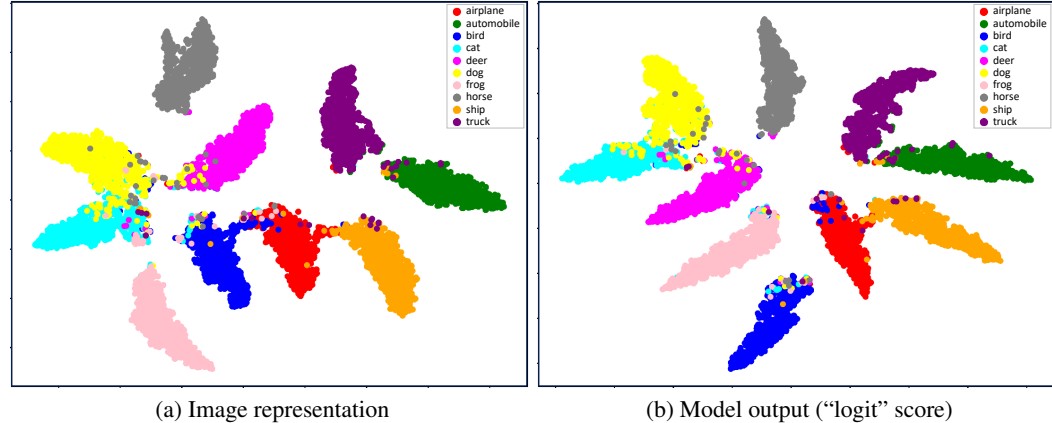

(a) Image representation          (b) Model output ("logit" score)

Figure 5: t-SNE visualization of the image representations and model outputs on CIFAR-10 test set. The model was trained for 128 epochs with 4000 labels. The same color means the same class.

As illustrated in Figure 5b, the model outputs of the samples from the same class are clustered relatively well. As a result, the image representations of the same-class samples are also clustered relatively well, as shown in Figure 5a. Consequently, by forcing the model outputs of the same-class samples to be as similar as possible, we obtain the similar image representations as well.

## B  COMPARISON OF METHODS

As presented in Section 4, we evaluate our method against four methods: MixMatch (Berthelot et al., 2019b), RealMix (Nair et al., 2019), ReMixMatch (Berthelot et al., 2019a), and FixMatch (Sohn et al., 2020). The comparison of the methods is shown in Table 6. RankingMatch[BA], RankingMatch[BH], RankingMatch[BM], and RankingMatch[CT] refer to RankingMatch with using BatchAll Triplet, BatchHard Triplet, BatchMean Triplet, and Contrastive loss respectively.

Table 6: Comparison of methods.

| Method | Data augmentation | Pseudo-label post-processing | Ranking loss | Note |
|---|---|---|---|---|
| MixMatch | Weak | Sharpening | None | Uses squared $L_2$ loss for unlabeled data |
| RealMix | Weak & Strong | Sharpening & Confidence threshold | None | Uses training signal annealing (TSA) to avoid overfitting |
| ReMixMatch | Weak & Strong | Sharpening | None | Uses extra rotation loss for unlabeled data |
| FixMatch | Weak & Strong | Confidence threshold | None | |
| RankingMatch[BA] | Weak & Strong | Confidence threshold | BatchAll Triplet loss | |
| RankingMatch[BH] | Weak & Strong | Confidence threshold | BatchHard Triplet loss | |
| RankingMatch[BM] | Weak & Strong | Confidence threshold | BatchMean Triplet loss | |
| RankingMatch[CT] | Weak & Strong | Confidence threshold | Contrastive loss | |

# C RankingMatch Algorithm

The full algorithm of RankingMatch is provided in Algorithm 1. Note that the meaning of $a$, $p$, $n$, $y_a$, $y_p$, $y_n$, $d_{a,p}$, $d_{a,n}$, and $f(\bullet)$ in Algorithm 1 were defined in Section 3.3. Algorithm 1 illustrates the use of BatchMean Triplet loss. When using Contrastive loss, most of the parts of Algorithm 1 are kept unchanged, except that $\mathcal{L}_s^{BM}$ and $\mathcal{L}_u^{BM}$ are replaced by Contrastive losses as presented in Section 3.3.2.

---

**Algorithm 1:** RankingMatch algorithm for computing loss function with using BatchMean Triplet loss.

---

**Input:** Batch of $B$ labeled samples and their one-hot labels $\mathcal{X} = \{(x_b, l_b) : b \in (1, ..., B)\}$, batch of $\mu B$ unlabeled samples $\mathcal{U} = \{u_b : b \in (1, ..., \mu B)\}$, confidence threshold $\tau$, margin $m$, and loss weights $\lambda_u$ and $\lambda_r$.

1   `/* ==================== Compute loss elements for labeled data ==================== */`

2   **for** $b = 1$ **to** $B$ **do**

3     $\hat{x}_b = p_{\mathrm{model}}(y \mid \mathcal{A}_w(x_b); \theta)$   `// "logits" score for weakly-augmented labeled data`

4   **end**

5   $\mathcal{L}_s^{CE} = \frac{1}{B} \sum_{b=1}^{B} \mathrm{H}(l_b, \mathrm{Softmax}(\hat{x}_b))$   `// Cross-Entropy loss for labeled data`

6   $\hat{\mathcal{X}} = \{\mathrm{L2Norm}(\hat{x}_b) : b \in (1, ..., B)\}$   `// Batch of B L₂-normalized "logits" scores for weakly-augmented labeled data`

7   $\mathcal{L}_s^{BM} = \frac{1}{|\hat{\mathcal{X}}|} \sum_{a \in \hat{\mathcal{X}}} f(m + \frac{1}{|\hat{\mathcal{X}}|} \sum_{\substack{p \in \hat{\mathcal{X}} \\ y_p = y_a}} d_{a,p} - \frac{1}{|\hat{\mathcal{X}}|} \sum_{\substack{n \in \hat{\mathcal{X}} \\ y_n \neq y_a}} d_{a,n})$   `// BatchMean Triplet loss for labeled data`

8   `/* ==================== Compute loss elements for unlabeled data ==================== */`

9   **for** $b = 1$ **to** $\mu B$ **do**

10     $q_b = p_{\mathrm{model}}(y \mid \mathcal{A}_w(u_b); \theta)$   `// "logits" score for weakly-augmented unlabeled data`

11     $\tilde{q}_b = \mathrm{Softmax}(q_b)$   `// Model prediction for weakly-augmented unlabeled data`

12     $\hat{q}_b = \mathrm{argmax}(\tilde{q}_b)$   `// One-hot pseudo-label for strongly-augmented unlabeled data`

13     $\hat{u}_b = p_{\mathrm{model}}(y \mid \mathcal{A}_s(u_b); \theta)$   `// "logits" score for strongly-augmented unlabeled data`

14   **end**

15   $\mathcal{L}_u^{CE} = \frac{1}{\mu B} \sum_{b=1}^{\mu B} \mathbb{1}(\max(\tilde{q}_b) \geq \tau) \, \mathrm{H}(\hat{q}_b, \mathrm{Softmax}(\hat{u}_b))$   `// Cross-Entropy loss for unlabeled data`

16   $\hat{\mathcal{U}} = \{\mathrm{L2Norm}(\hat{u}_b) : b \in (1, ..., \mu B)\}$   `// Batch of μB L₂-normalized "logits" scores for strongly-augmented unlabeled data`

17   $\mathcal{L}_u^{BM} = \frac{1}{|\hat{\mathcal{U}}|} \sum_{a \in \hat{\mathcal{U}}} f(m + \frac{1}{|\hat{\mathcal{U}}|} \sum_{\substack{p \in \hat{\mathcal{U}} \\ y_p = y_a}} d_{a,p} - \frac{1}{|\hat{\mathcal{U}}|} \sum_{\substack{n \in \hat{\mathcal{U}} \\ y_n \neq y_a}} d_{a,n})$   `// BatchMean Triplet loss for unlabeled data`

18   `/* ==================== Compute the total loss ==================== */`

19   $\mathcal{L} = \mathcal{L}_s^{CE} + \lambda_u \mathcal{L}_u^{CE} + \lambda_r (\mathcal{L}_s^{BM} + \mathcal{L}_u^{BM})$

20   **return** $\mathcal{L}$

---

# D Details of Training Protocol and Hyperparameters

## D.1 Optimizer and Learning Rate Schedule

We use the same codebase, data pre-processing, optimizer, and learning rate schedule for methods implemented by us. An SGD optimizer with momentum is used for training the models. Additionally, we apply a cosine learning rate decay (Loshchilov & Hutter, 2016) which effectively decays the learning rate by following a cosine curve. Given a base learning rate $\eta$, the learning rate at the training step $s$ is set to

$$\eta \cos\left(\frac{7\pi s}{16S}\right) \tag{8}$$

where $S$ is the total number of training steps.

Concretely, $S$ is equal to the number of epochs multiplied by the number of training steps within one epoch. Finally, we use Exponential Moving Average (EMA) to obtain the model for evaluation.

## D.2 LIST OF HYPERPARAMETERS

For all our experiments, we use

- A batch size of 64 for all datasets except that STL-10 uses a batch size of 32,
- Nesterov Momentum with a momentum of 0.9,
- A weight decay of 0.0005 and a base learning rate of 0.03.

For other hyperparameters, we first define notations as in Table 7.

Table 7: Hyperparameter definition.

| Notation | Definition |
|----------|------------|
| $T_{mix}$ | Temperature parameter for sharpening used in MixMatch |
| $K$ | Number of augmentations used when guessing labels in MixMatch |
| $\alpha$ | Hyperparameter for the Beta distribution used in MixMatch |
| $\tau$ | Confidence threshold used in FixMatch and RankingMatch for choosing high-confidence predictions |
| $m$ | Margin used in RankingMatch with using Triplet loss |
| $T$ | Temperature parameter used in RankingMatch with using Contrastive loss |
| $\lambda_u$ | A hyperparameter weighting the contribution of the unlabeled examples to the training loss. In RankingMatch, $\lambda_u$ is the weight determining the contribution of Cross-Entropy loss of unlabeled data to the overall loss. |
| $\lambda_r$ | A hyperparameter used in RankingMatch to determine the contribution of Ranking loss element to the overall loss |

The details of hyperparameters for all methods are shown in Table 8.

Table 8: Details of hyperparameters. As presented in Section 4.1, FixMatch[RA] refers to FixMatch with using RandAugment; RankingMatch[BM] and RankingMatch[CT] refer to RankingMatch with using BatchMean Triplet loss and Contrastive loss respectively.

| Method | $\lambda_u$ | $\lambda_r$ | $T_{mix}$ | $K$ | $\alpha$ | $\tau$ | $m$ | soft-margin | $T$ | $L_2$-normalization |
|--------|-------------|-------------|-----------|-----|----------|--------|-----|-------------|-----|---------------------|
| MixMatch | 75 | - | 0.5 | 2 | 0.75 | - | - | - | - | - |
| FixMatch[RA] | 1 | - | - | - | - | 0.95 | - | - | - | - |
| RankingMatch[BM] | 1 | 1 | - | - | - | 0.95 | 0.5 | True | - | True |
| RankingMatch[CT] | 1 | 1 | - | - | - | 0.95 | - | - | 0.2 | True |

## D.3 AUGMENTATION DETAILS

**For weak augmentation**, we adopt standard padding-and-cropping and horizontal flipping augmentation strategies. We set the padding to 4 for CIFAR-10, CIFAR-100, and SVHN. Because STL-10

and Tiny ImageNet have larger image sizes, a padding of 12 and 8 is used for STL-10 and Tiny ImageNet, respectively. Notably, we did not apply horizontal flipping for the SVHN dataset.

**For strong augmentation**, we first randomly pick 2 out of 14 transformations. These 14 transformations consist of Autocontrast, Brightness, Color, Contrast, Equalize, Identity, Posterize, Rotate, Sharpness, ShearX, ShearY, Solarize, TranslateX, and TranslateY. Then, Cutout (DeVries & Taylor, 2017) is followed to obtain the final strongly-augmented sample. We set the cutout size to 16 for CIFAR-10, CIFAR-100, and SVHN. A cutout size of 48 and 32 is used for STL-10 and Tiny ImageNet, respectively. For more details about 14 transformations used for strong augmentation, readers could refer to FixMatch (Sohn et al., 2020).

## D.4 DATASET DETAILS

**CIFAR-10 and CIFAR-100** are widely used datasets that consist of $32 \times 32$ color images. Each dataset contains 50000 training images and 10000 test images. Following standard practice, as mentioned in Oliver et al. (2018), we divide training images into train and validation split, with 45000 images for training and 5000 images for validation. Validation split is used for hyperparameter tuning and model selection. In train split, we discard all except a number of labels (40, 250, and 4000 labels for CIFAR-10; 400, 2500, and 10000 labels for CIFAR-100) to vary the labeled data set size.

**SVHN** is a real-world dataset containing 73257 training images and 26032 test images. We use the similar data strategy as used for CIFAR-10 and CIFAR-100. We divide training images into train and validation split, with 65937 images used for training and 7320 images used for validation. In train split, we discard all except a number of labels (40, 250, and 1000 labels) to vary the labeled data set size.

**STL-10** is a dataset designed for unsupervised learning, containing 5000 labeled training images and 100000 unlabeled images. There are ten pre-defined folds with 1000 labeled images each. Given a fold with 1000 labeled images, we use 4000 other labeled images out of 5000 labeled training images as validation split. The STL-10 test set has 8000 labeled images.

**Tiny ImageNet** is a compact version of ImageNet, including 100000 training images, 10000 validation images, and 10000 test images. Since the ground-truth labels of test images are not available, we evaluate our method on 10000 validation images and use them as the test set. There are 200 classes in Tiny ImageNet. We divide training images into 90000 images used for train split and 10000 used for validation split. For the semi-supervised learning setting, we use 10% of train split as labeled data and treat the rest as unlabeled data. As a result, there are 9000 labeled images and 81000 unlabeled images.

# E  QUALITATIVE RESULTS

## E.1  RANKINGMATCH VERSUS OTHER METHODS

To cast the light for how the models have learned to classify the images, we visualize the "logits" scores using t-SNE which was introduced by Maaten & Hinton (2008). t-SNE visualization reduces the high-dimensional features to a reasonable dimension to help grasp the tendency of the learned models. We visualize the "logits" scores of four methods, which are MixMatch, FixMatch[RA], RankingMatch[BM], and RankingMatch[CT], as shown in Figure 6. These four methods were trained on CIFAR-10 with 4000 labels and were trained for 128 epochs with the same random seed.

At first glance in Figure 6, both four methods tend to group the points of the same class into the same cluster depicted by the same color. The shape of the clusters is different among methods, and it is hard to say which method is the best one based on the shape of the clusters. However, the less the overlapping points among classes are, the better the method is. We can easily see that MixMatch (Figure 6a) has more overlapping points than other methods, leading to worse performance. This statement is consistent with the accuracy of the method. We quantify the overlapping points by computing the confusion matrices, as shown in Figure 7.

If we pay more attention to t-SNE visualization in Figure 6, we can realize that all methods have many overlapping points between class 3 (*cat*) and 5 (*dog*). These overlapping points could be regarded as the confusion points, where the model misclassifies them. For example, as shown in the

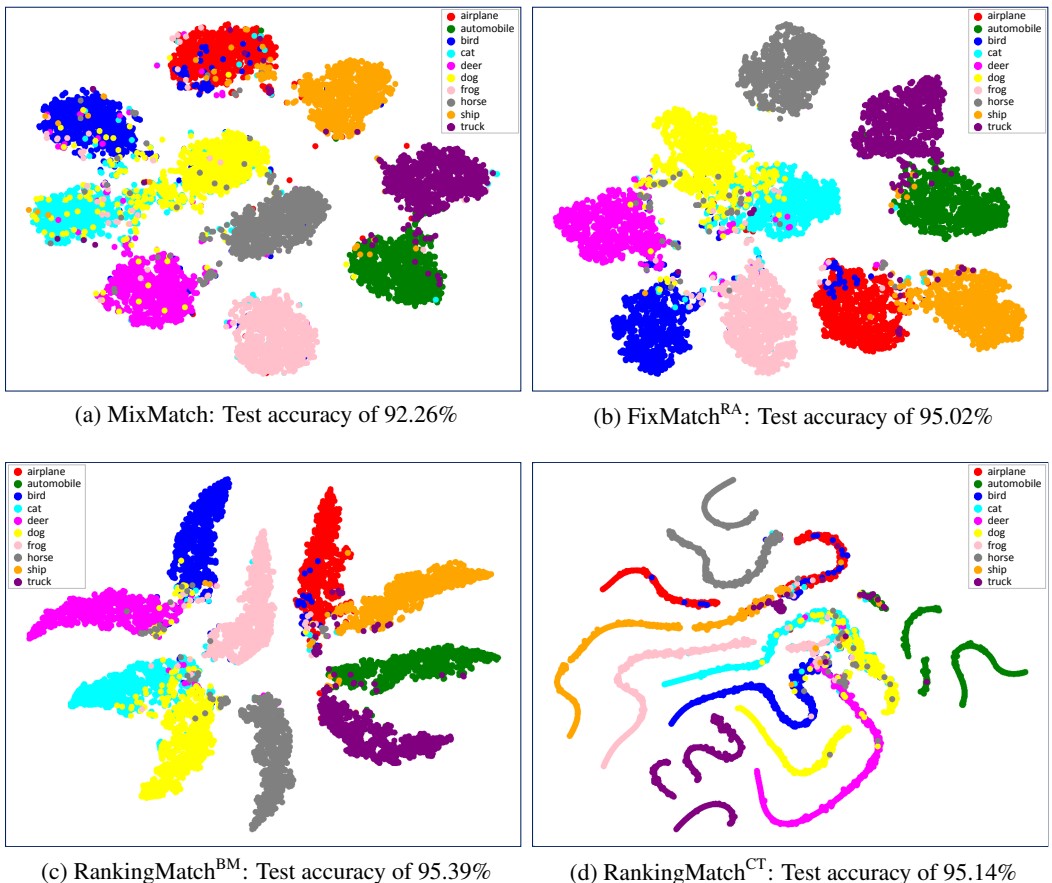

(a) MixMatch: Test accuracy of 92.26%

(b) FixMatch[RA]: Test accuracy of 95.02%

(c) RankingMatch[BM]: Test accuracy of 95.39%

(d) RankingMatch[CT]: Test accuracy of 95.14%

Figure 6: t-SNE visualization of the "logits" scores of the methods on CIFAR-10 test set. The models were trained for 128 epochs with 4000 labels. Note that this figure contains higher-resolution versions of the figures shown in Figure 2.

confusion matrices in Figure 7, MixMatch misclassifies 100 points as *dog* while they are actually *cat*. This number is 66, 60, or 64 in the case of FixMatch[RA], RankingMatch[BM], or RankingMatch[CT], respectively. We leave researching the shape of the clusters and the relationship between t-SNE visualization and the confusion matrix for future work.

### E.2 RANKINGMATCH WITH VARIANTS OF TRIPLET LOSS

Figure 8 shows t-SNE visualization for the "logits" scores of the models in Table 5 in the case of trained on CIFAR-10 with 4000 labels. Triplet loss utilizes a series of triplets $\{a, p, n\}$ to satisfy the objective function. Once the input was given, the loss function is optimized to minimize the distance between $a$ and $p$ while maximizing the distance between $a$ and $n$, implying that the way of treating the series of triplets might significantly affect how the model is updated. BatchAll, for instance, takes into account all possible triplets when calculating the loss function. Since BatchAll treats all samples equally, it is likely to be biased by the samples with predominant features, which might hurt expected performance. To shore up our argument, let see in Figure 8a, BatchAll has numerous overlapping points and even gets lower accuracy by a large margin compared to others. Especially at the center of the figure, the model is confusing almost all the labels. It is thus natural to argue that BatchAll is poor at generalizing to unseen data. BatchHard (Figure 8b) is better than BatchAll, but it still has many overlapping points at the center of the figure. Our BatchMean surpasses both BatchHard and BatchAll when much better clustering classes, leading to the best accuracy compared

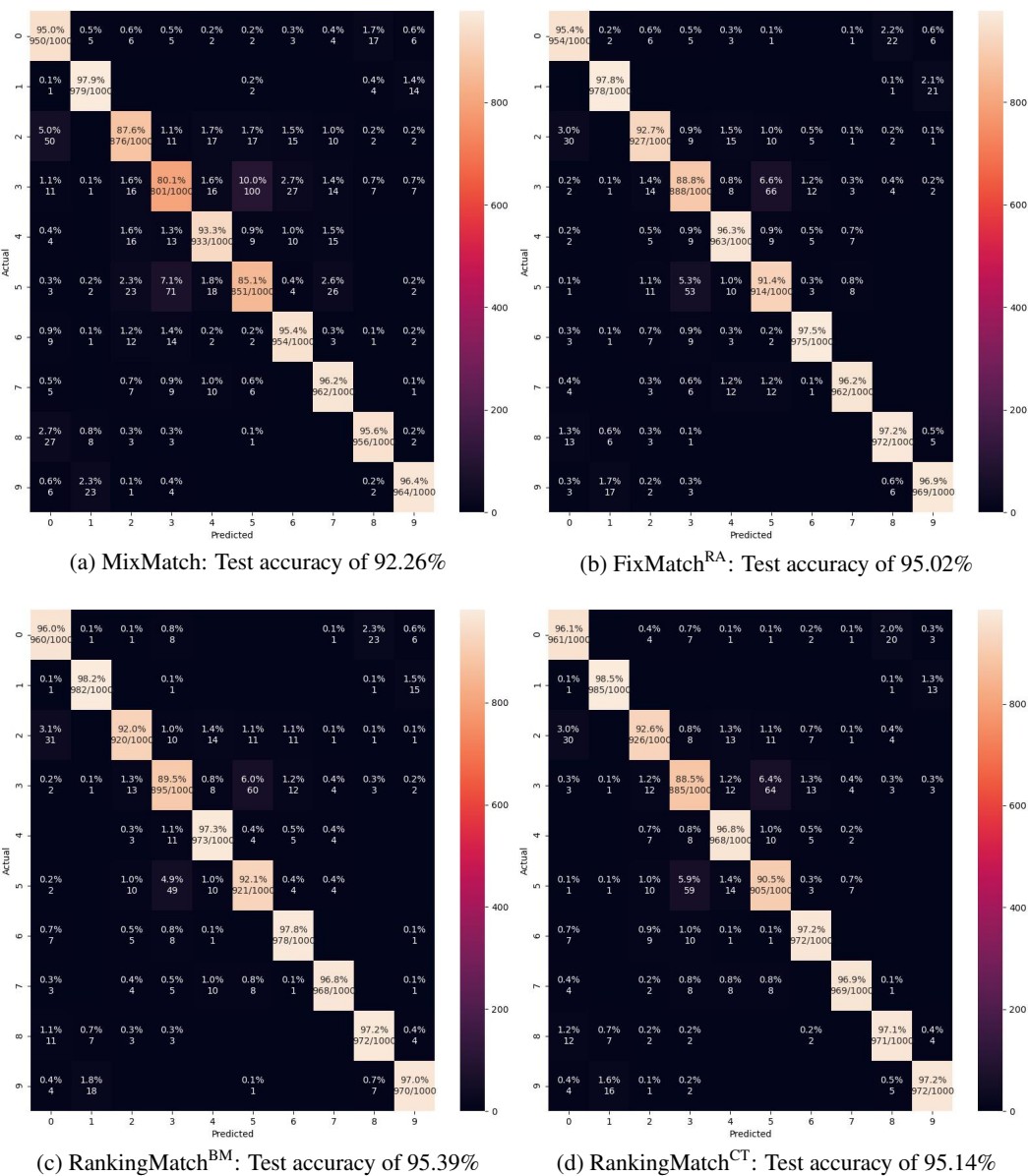

Figure 7: Confusion matrices for models in Figure 6. Classes in Figure 6 are numbered from 0 to 9, respectively.

to other methods. The confusion matrices shown in Figure 9 quantify overlapping points, which could be regarded as confusion points where the model misclassifies them.

### E.3 RANKINGMATCH WITH $L_2$-NORMALIZATION

We use the models reported in Table 5 in the case of trained on CIFAR-10 with 4000 labels. Notably, we do not visualize RankingMatch[BM] without $L_2$-normalization because that model does not converge. t-SNE visualizations of the "logits" scores of RankingMatch[CT] models and corresponding confusion matrices are shown in Figure 10 and 11, respectively. There is not much difference between RankingMatch[CT] with and without $L_2$-normalization in terms of the cluster shape and overlapping points. However, in terms of accuracy, $L_2$-normalization actually helps improve classification performance, as shown in Table 5.

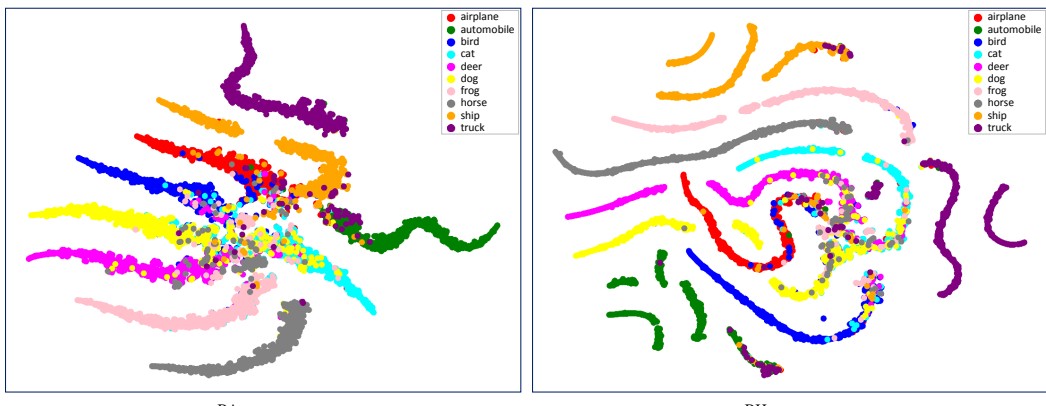

(a) RankingMatch[BA]: Test accuracy of 87.95%    (b) RankingMatch[BH]: Test accuracy of 91.41%

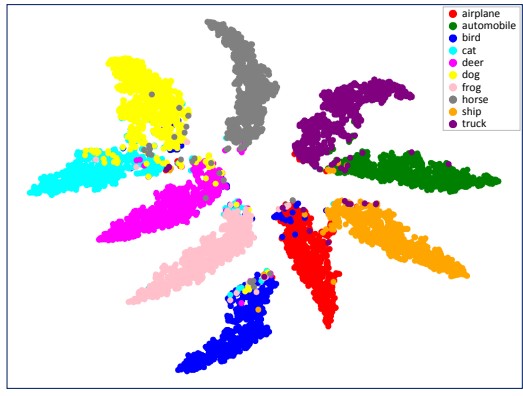

(c) RankingMatch[BM]: Test accuracy of 95.51%

Figure 8: t-SNE visualization of the "logits" scores of RankingMatch with variants of Triplet loss on CIFAR-10 test set. The models were trained for 128 epochs with 4000 labels.

## F    COMPUTATIONAL EFFICIENCY OF BATCHMEAN TRIPLET LOSS

As presented in Section 3.3,

- BatchAll Triplet loss considers all possible triplets when computing the loss.
- BatchHard Triplet loss only takes into account the hardest triplets when calculating the loss.
- Our BatchMean Triplet loss only considers the "mean" triplets (consisting of anchors, "mean" positive samples, and "mean" negative samples) when computing the loss.

Because BatchMean does not consider all triplets but only the "mean" triplets, BatchMean has the advantage of computational efficiency of BatchHard Triplet loss. On the other hand, all samples are used to compute the "mean" samples, BatchMean also takes into account all samples as done in BatchAll Triplet loss. The efficacy of BatchMean Triplet loss was proved in Table 5 when achieving the lowest error rates compared to other methods. Therefore, this section only focuses on the contents of computational efficiency. Firstly, let us take a simple example to intuitively show the computational efficiency of BatchHard and BatchMean against BatchAll Triplet loss. Assume we have an anchor $a$, three positive samples corresponding to $a$: $p_1$, $p_2$, and $p_3$, and two negative samples with respect to $a$: $n_1$ and $n_2$.

- In BatchAll, there will have six possible triplets considered: $(a, p_1, n_1)$, $(a, p_1, n_2)$, $(a, p_2, n_1)$, $(a, p_2, n_2)$, $(a, p_3, n_1)$, and $(a, p_3, n_2)$.

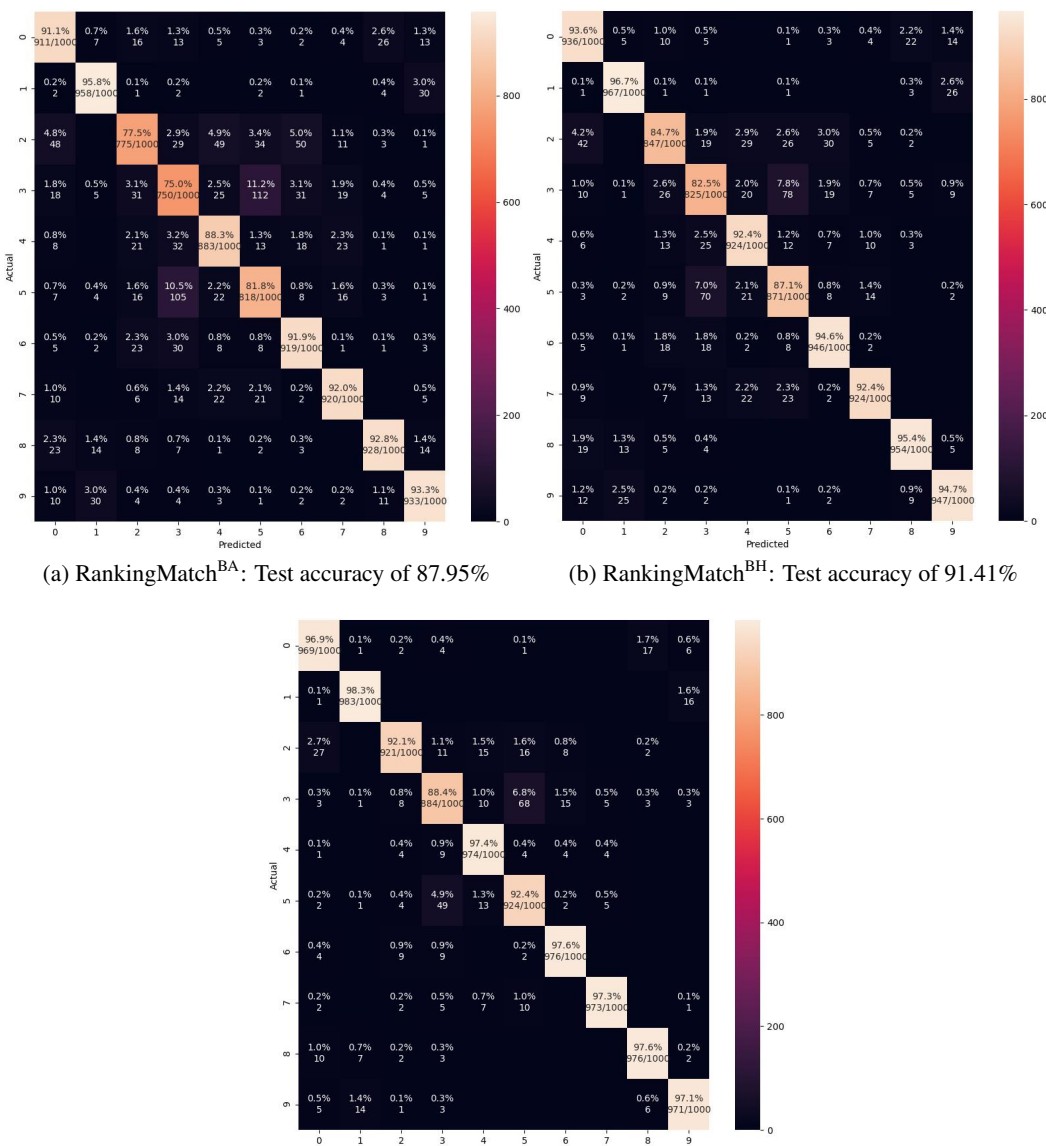

(a) RankingMatch$^{BA}$: Test accuracy of 87.95%   (b) RankingMatch$^{BH}$: Test accuracy of 91.41%

(c) RankingMatch$^{BM}$: Test accuracy of 95.51%

Figure 9: Confusion matrices for models in Figure 8. Classes in Figure 8 are numbered from 0 to 9, respectively.

- BatchHard only takes into account one hardest triplet: $(a, \text{furthest}(p_1, p_2, p_3), \text{nearest}(n_1, n_2))$.

- Finally, in our BatchMean, there is only one "mean" triplet considered: $(a, \text{mean}(p_1, p_2, p_3), \text{mean}(n_1, n_2))$.

As a result, BatchHard and BatchMean take fewer computations than BatchAll Triplet loss.

To quantitatively prove the computational efficiency of BatchHard and our BatchMean compared to BatchAll Triplet loss, we measure the training time and GPU memory usage, as presented in Appendix F.1 and F.2. We use the same hyperparameters for all methods to ensure a fair comparison. Notably, for clearance and simplicity, we use BatchAll, BatchHard, and BatchMean for RankingMatch$^{BA}$, RankingMatch$^{BH}$, and RankingMatch$^{BM}$ respectively.

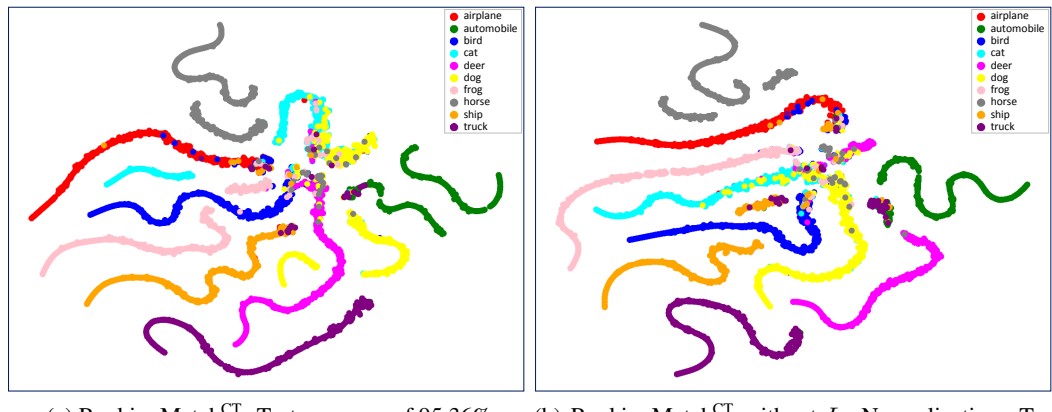

(a) RankingMatch$^{CT}$: Test accuracy of 95.36%

(b) RankingMatch$^{CT}$ without $L_2$-Normalization: Test accuracy of 95.33%

Figure 10: t-SNE visualization of the "logits" scores of RankingMatch with Contrastive loss on CIFAR-10 test set. The models were trained for 128 epochs with 4000 labels.

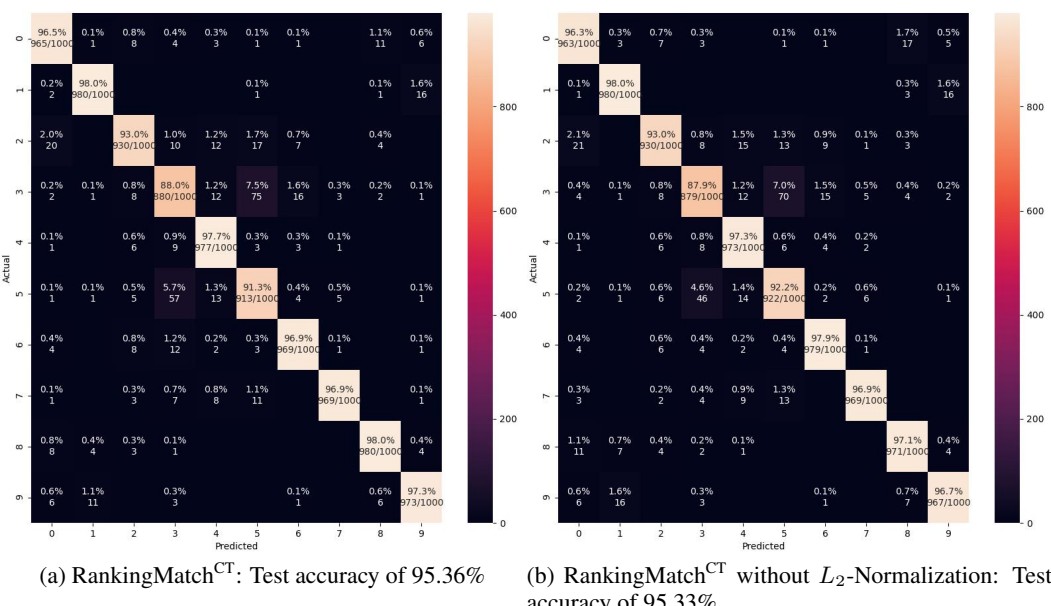

(a) RankingMatch$^{CT}$: Test accuracy of 95.36%

(b) RankingMatch$^{CT}$ without $L_2$-Normalization: Test accuracy of 95.33%

Figure 11: Confusion matrices for models in Figure 10. Classes in Figure 10 are numbered from 0 to 9, respectively.

## F.1 MEASUREMENT PER EPOCH

Table 9 shows the training time per epoch (seconds) and GPU memory usage (MB) of the methods for 128 epochs on CIFAR-10, SVHN, and CIFAR-100. As shown in Table 9, BatchHard and our BatchMean have the similar training time and the similar GPU memory usage among datasets. The results also show that BatchHard and BatchMean are more computationally efficient than BatchAll across all datasets. For example:

- On SVHN, BatchHard and BatchMean reduce the training time per epoch by 126.25 and 125.82 seconds compared to BatchAll, respectively.

- BatchAll occupies much more GPU memory than BatchHard and BatchMean, which is about 1.87, 1.85, and 1.79 times on CIFAR-10, SVHN, and CIFAR-100 respectively.

Table 9: Training time per epoch (seconds) and GPU memory usage (MB) for 128 epochs on CIFAR-10, SVHN, and CIFAR-100.

| | Training time per epoch | | |
| --- | --- | --- | --- |
| Method | CIFAR-10 | SVHN | CIFAR-100 |
| BatchAll | 385.87±15.35 | 434.05±20.07 | 323.03±8.81 |
| BatchHard | **308.99±0.79** | **307.80±0.71** | **310.16±0.98** |
| BatchMean | **309.27±0.74** | **308.23±0.57** | **309.33±1.03** |
| | GPU memory usage | | |
| Method | CIFAR-10 | SVHN | CIFAR-100 |
| BatchAll | 9039.72±2043.30 | 8967.59±1535.99 | 8655.81±2299.36 |
| BatchHard | **4845.92±0.72** | **4845.92±0.72** | **4847.86±0.97** |
| BatchMean | **4845.92±0.72** | **4845.92±0.72** | **4847.86±0.97** |

The training time per epoch (seconds) and GPU memory usage (MB) are measured during 128 epochs, as illustrated in Figure 12. In addition to computational efficiency against BatchAll, BatchHard and BatchMean have the stable training time per epoch and the stable GPU memory usage. On the other hand, the training time of BatchAll is gradually increased during the training process. Especially, there is a time when the training time of BatchAll grows up significantly, and this time is different among datasets. Moreover, it seems that the amount of computations of BatchAll is also different among datasets. These differences will be clarified in the following section (Appendix F.2).

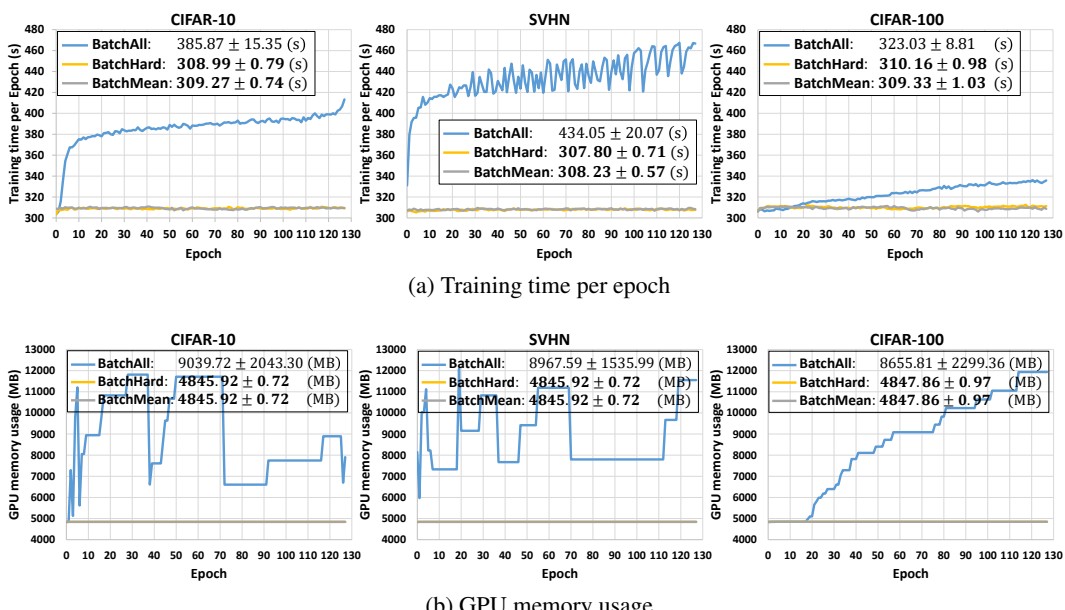

(a) Training time per epoch

(b) GPU memory usage

Figure 12: Training time per epoch (seconds) and GPU memory usage (MB) during 128 epochs on CIFAR-10, SVHN, and CIFAR-100.

## F.2 MEASUREMENT PER BATCH

Different from the previous section (Appendix F.1), this section measures the training time and monitors the GPU memory usage per training step. Each training step corresponds to the training

over a batch. Table 10 and Figure 13 show the measurement of the methods for the first 5100 training steps on CIFAR-10 and SVHN.

Table 10: Training time per batch (milliseconds) and GPU memory usage (MB) for the first 5100 training steps on CIFAR-10 and SVHN.

| Method | CIFAR-10 | | SVHN | |
|---|---|---|---|---|
| | Time per batch | GPU memory usage | Time per batch | GPU memory usage |
| BatchAll | 318.48±21.02 | 6643.71±2222.58 | 371.04±28.64 | 8974.89±2167.56 |
| BatchHard | **300.97±7.28** | **4844.00±3.10** | **302.21±7.77** | **4843.97±3.11** |
| BatchMean | **302.37±7.92** | **4843.98±3.11** | **303.18±7.87** | **4843.95±3.12** |

Table 10 demonstrates that BatchHard and our BatchMean is much more computationally efficient than BatchAll. For instance, on SVHN, BatchHard and BatchMean reduce the training time per batch by 68.83 and 67.86 milliseconds compared to BatchAll, respectively. It would be found more interesting in Figure 13.

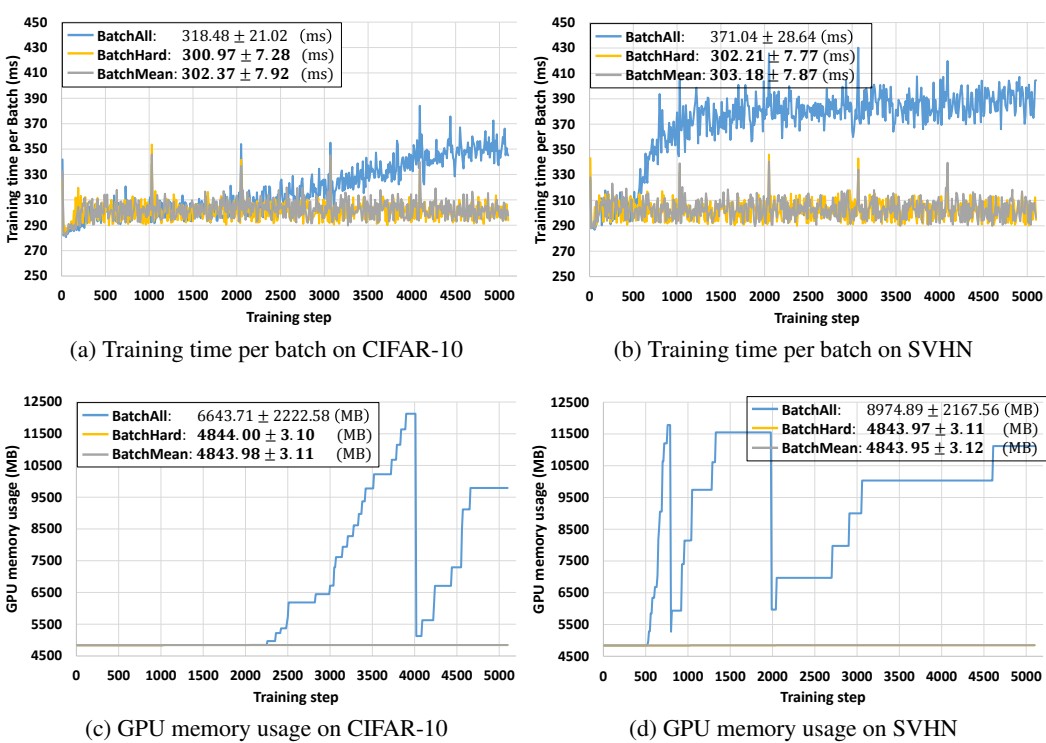

(a) Training time per batch on CIFAR-10

(b) Training time per batch on SVHN

(c) GPU memory usage on CIFAR-10

(d) GPU memory usage on SVHN

Figure 13: Training time per batch (milliseconds) and GPU memory usage (MB) during the first 5100 training steps on CIFAR-10 and SVHN.

In Figure 13a and 13b, the "peak" values indicate the starting time of a new epoch. At that time, there are some extra steps like initialization, so it might take more time. As shown in Figure 13, BatchAll starts to take more computations from the $2200^{th}$ and $500^{th}$ training step on CIFAR-10 and SVHN, respectively; this is reasonable because we used a threshold to ignore the low-confidence predictions for unlabeled data (Section 3.2). At the beginning of the training process, the model is not well trained and thus produces the predictions with very low confidence, so many samples are discarded. As a result, there are a few possible triplets for unlabeled data at the beginning of the training process, leading to fewer computations of BatchAll.

When the model is progressed, it is trained more and produces more high-confidence predictions, leading to more possible triplets. Therefore, BatchAll has more computations. Figure 13 also shows

that the starting point of increasing the computation of BatchAll is earlier in the case of SVHN compared to CIFAR-10. This is reasonable because the SVHN dataset only consists of digits from 0 to 9 and thus is simpler than the CIFAR-10 dataset. As a result, it is easier for the model to learn SVHN than CIFAR-10, leading to more high-confidence predictions and more possible triplets at the beginning of the training process in the case of SVHN compared to CIFAR-10. Moreover, the training time per batch and GPU memory usage of BatchAll on SVHN are larger than those on CIFAR-10 over the first 5100 training steps. Therefore, we can argue that *the less complex the dataset is, the earlier and more BatchAll takes computations*. This is also the reason for us to monitor the computational efficiency with more training steps on CIFAR-100.

Since CIFAR-100 has 100 classes, it is more complex than CIFAR-10 and SVHN. Therefore, the model needs more training steps to be more confident. Table 11 and Figure 14 show the training time per batch and GPU memory usage of the methods on CIFAR-100 over the first 31620 training steps. BatchHard and our BatchMean are still more computationally efficient than BatchAll, but it is not too much especially for the training time per batch. To show discernible changes, we need to monitor the training time per batch and GPU memory usage with more training steps, as presented in the previous section (Appendix F.1). Figure 14 also shows that BatchAll starts to consume more computations at around the $17500^{th}$ training step, which is much later than on CIFAR-10 and SVHN.

Table 11: Training time per batch (milliseconds) and GPU memory usage (MB) for the first 31620 training steps on CIFAR-100.

| Method | Time per batch | GPU memory Usage |
|---|---|---|
| BatchAll | 306.32±8.48 | 5255.91±593.29 |
| BatchHard | **303.29±7.67** | **4846.80±2.58** |
| BatchMean | **302.86±7.59** | **4846.77±2.57** |

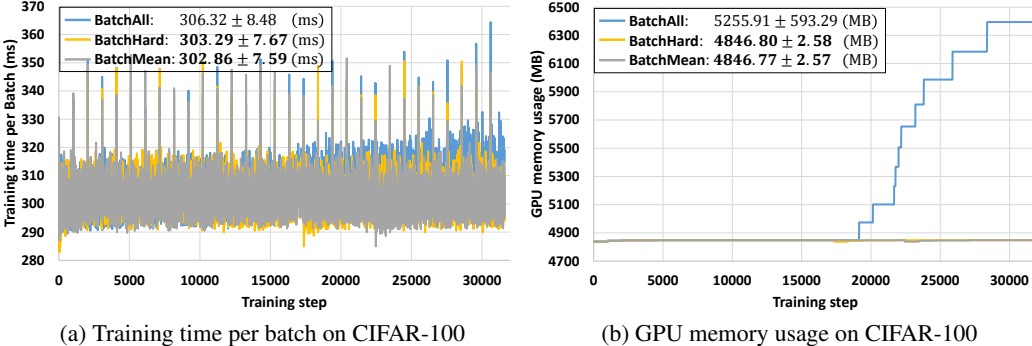

(a) Training time per batch on CIFAR-100  (b) GPU memory usage on CIFAR-100

Figure 14: Training time per batch (milliseconds) and GPU memory usage (MB) during the first 31620 training steps on CIFAR-100.

