# OpenReview forum: "RankingMatch: Delving into Semi-Supervised Learning with Consistency Regularization and Ranking Loss"
_ICLR.cc/2021/Conference — Reject_

### Official Review · AnonReviewer4 · 2020-10-24
**interesting work; more thorough experiments will be helpful**

**Rating:** 6
**Confidence:** 4

**Review:**

********Summary
In this paper, they introduced a new objective function, dubbed BatchMean Triplet loss, for Semi-supervised learning on image classification problems. In this framework both labeled and unlabeled data are used for training at the same time. The labeled data will be augmented weakly and cross entropy loss as well as ranking loss will be computed on their output. The unlabeled data will be augmented weakly and strongly, and the label based on the weakly augmented will be a pseudo label for the strongly augmented input, and similarly cross entropy and rank loss will be computed for the unlabeled data. For the rank loss, they used Contrastive loss and BatchMean Triplet loss, which is a variant of Triplet loss. They applied Triplet and Contrastive loss to the model output instead of image representation.

********Positives
- The paper is well-written and well-organized. Figure 1 is self-explanatory, showing the general framework introduced in this paper.

- They covered background and related works to this paper when explaining their method, which makes the work clear.

********Notes
- My main concern about this work is lack of contribution. This could be negligible if I see a better result in their experiments section. Table 1, their performance is very similar to FixMatch(RA). I do not see much improvement of their method over baselines in Table 2 (they are on CIFAR-10 and CIFAR-100 only).

- They have mentioned that "Our BatchMean Triplet loss has the advantage of computational efficiency of existing BatchHard Triplet loss while taking into account all input samples." It would be interesting to quantitatively see how much their method is computational efficient.

- It would be interesting to see the comparison of two ranking losses explained here (BatchMean Triplet loss vs Contrastive loss).

- I would like to see the comparison of their method against baselines on all of the datasets explained here. I understand the limited computing resources issue, that is really unfortunate that some institutes cannot provide enough computational resources for their researchers experiments. But as a reviewer, I need to see how this method works on a wider range of datasets.

********Reason to accept or reject
The paper is well-written and they explained their method clearly, which is important. My concern is lack of contribution. If they could provide more thorough experiments, this could be more helpful work for other researchers in this area.

---

> ### Author Response · Authors · 2020-11-17
> **[Author's response] Thank you so much for the constructive feedback!**
>
> Dear reviewer,
>
> First of all, we would like to thank you so much for the positive and constructive feedback. We respond to your four concerns mentioned in the “notes”, as follows:
> 1. Lack of contribution: RankingMatch’s performance is very similar to FixMatch’s performance:
> - In the original paper, **FixMatch** was trained for **1024 epochs** to obtain the state-of-the-art results. However, our **RankingMatch** was only trained for **128** and **256 epochs** to achieve the state-of-the-art results, leading to a **faster training time** compared to FixMatch. Concretely, RankingMatch obtains the state-of-the-art results on CIFAR-10 with 250 labels (Table 2), CIFAR-100 with 10000 labels (Table 2), and SVHN with 250 and 1000 labels (Table 3) (*256 epochs for CIFAR-10 and 128 epochs for other datasets*).
> - We also show the results when FixMatch is trained for **128 epochs** (Table 1, 3, and 4). When trained for 128 epochs, RankingMatch ($RankingMatch^{BM}$ or $RankingMatch^{CT}$) obtains the better results compared to FixMatch. Especially, both $RankingMatch^{BM}$ and $RankingMatch^{CT}$ outperform FixMatch on CIFAR-10, as shown in Table 1. For example, $RankingMatch^{BM}$ and $RankingMatch^{CT}$ reduce the error rate by 4.20% and 2.76% compared to $Fixmatch^{RA}$ on CIFAR-10 with 40 labels, respectively (Table 1).
> 2. Quantitatively show how much BatchMean Triplet loss is computationally efficient:
> - Since we could not insert the figures to Openreview’s response and the number of characters is limited, we provide a **detailed explanation** for how much our BatchMean Triplet loss is computational efficient at:
> https://drive.google.com/file/d/1yku_1Pt9f5Cacw_jz08NEFtWLg9GJBAp/view?usp=sharing
> 3. Comparison of BatchMean Triplet loss and Contrastive loss:
> - Both BatchMean Triplet loss and Contrastive loss measure the similarity among samples, but these two losses have different measurements. BatchMean Triplet loss uses Euclidean distance while Contrastive loss utilizes cosine similarity.
> - Due to the different measurements, BatchMean Triplet loss and Contrastive loss also have different working principles. Firstly, let positive samples be the samples from the same class and negative samples be the samples from the different classes. In our paper, **the samples refer to the “logits” scores**.
> In BatchMean Triplet loss, given a sample referred to as an anchor, the Euclidean distance between the anchor and its negative sample is encouraged to be larger than the Euclidean distance between the anchor and its positive sample by at least a margin *m*.
> On the other hand, in Contrastive loss, the cosine similarity of positive samples is maximized, while the cosine similarity of negative samples is minimized.
> - We experimented and quantitatively showed that cosine similarity becomes **mostly better** than Euclidean distance when **the dimension** of the samples **is increased**, leading to a better performance of Contrastive loss compared to BatchMean Triplet loss. For instance, on CIFAR-10 where the “logits” scores are **10-dimensional** vectors, **$RankingMatch^{BM}$ is better than $RankingMatch^{CT}$** in all cases (Table 1 and 2). However, on CIFAR-100 where the “logits” scores are **100-dimensional vectors**, **$RankingMatch^{CT}$ becomes better than $RankingMatch^{BM}$** in all cases except the case of 400 labels (Table 1 and Table 2).
> - It would be found interesting via **our visualization**, as shown in **Figure 2**. As shown in that figure, in the “logits” space, $RankingMatch^{BM}$ and $RankingMatch^{CT}$ have different shapes of the clusters, reflecting the difference in the working principle mentioned above.
> 4. How RankingMatch works on a wider range of datasets:
> - We would like to thank you for your understanding of our limited computing resources issue. Actually, we compared our method ($RankingMatch^{BM}$ and $RankingMatch^{CT}$) against four methods (MixMatch, RealMix, ReMixMatch, and FixMatch) on four datasets (CIFAR-10, CIFAR-100, SVHN, and STL-10). We did not run one time for each result but run **multiple times**, then taking the average and standard deviation. Hence, due to the limited computing resources, in addition to two versions of our method ($RankingMatch^{BM}$ and $RankingMatch^{CT}$), we could only re-implement and do our experiments for two methods. We chose re-implementing MixMatch and FixMatch because these two **original papers** also conducted experiments on CIFAR-10, CIFAR-100, SVHN, and STL-10. On the other hand, the RealMix paper did not do experiments on CIFAR-100 and STL-10, and the ReMixMatch paper did not conduct experiments on CIFAR-100.
> - However, to respond to your concern, we are **implementing** and **doing experiments** on another dataset, which is **Tiny-ImageNet**. We will report the results once it is completed.
>
> *To be continued in the following response due to the limited number of characters...*

---

> > ### Author Response · Authors · 2020-11-17
> > **[Author's response] Continued**
> >
> > To be concluded:
> > - We will add the quantitative comparisons of how much our BatchMean Triplet loss is computationally efficient to the paper.
> > - We will add the results on the Tiny-ImageNet dataset.
> > - We expect that our responses can satisfy your concerns and help consolidate our work. Based on that, we hope you can reevaluate our paper.
> >
> > Thank you so much for spending time reviewing our paper, and we are looking forward to hearing from you!

---

> ### Author Response · Authors · 2020-11-24
> **[Author's response] Paper and answers updated!**
>
> Dear reviewer,
>
> According to your concerns, our paper and answers are updated as follows:
> 1. We added the quantitative results of how much our BatchMean Triplet loss is computationally efficient to **Section 4.6** (**Analysis – Computational efficiency**). We also added more results and explanations to **Appendix F** (**Computational Efficiency of BatchMean Triplet loss**).
> 2. We added the results for the Tiny ImageNet dataset to **Section 4.4** (**Tiny ImageNet**) to prove that our method can work on a wider range of datasets.
>
> Thank you so much for reading, and we hope you can reconsider and reevaluate our paper!
>
> **Notably**, because our paper is updated, some figures have the number changed. Therefore, we updated the figure number in the previous response.

---

### Official Review · AnonReviewer3 · 2020-10-25
**No clear advantage over FixMatch**

**Rating:** 3
**Confidence:** 5

**Review:**

Summary:
The paper presents an SSL method extending FixMatch by introducing an auxiliary loss motivated from the metric learning literature. For example, the triplet loss is utilized to define the loss, where any triplet of anchor, positive and negative examples, either using ground-truth labels for labeled data or pseudo-labels for unlabeled data, are used to compute the loss. Variants using hard triplets of the mean as well as using the contrastive loss are proposed. The proposed methods are evaluated on standard SSL benchmarks.

Review comments:
While authors presented ablation study with various metric learning losses proposed in this work, the biggest concern is that none of these losses shows the clear performance improvement over FixMatch when combined with FixMatch losses. This begs the essential question on the effectiveness of the proposed losses on top of FixMatch. If RankingMatch is not clearly more performant than FixMatch, is there any scenario that FixMatch cannot be applied but RankingMatch can?

It is also unclear to me the motivation for introducing auxiliary metric learning losses -- why is it a good addition to the FixMatch loss? Which aspect of FixMatch loss is problematic and how does adding metric learning loss fix that issue?

"We argue that the images from the same class do not have to have similar representations strictly, but their model outputs should be as similar as possible" --> There seems no supporting empirical evidence to this statement.

Reason for decision:
Given the fact that the proposed method is constructed incrementally to the previous method (i.e., additional loss to FixMatch loss), the initial decision criteria is its empirical effectiveness. Unfortunately, I am not convinced that the proposed method is clearly improving upon previous methods in any settings considered in the paper, so I would recommend for rejection.

I have read author response and thank authors for their response. I have decided to keep my initial rating. I am not convinced by the response that the proposed method has a clear advantage over Fixmatch as some of their reported numbers for Fixmatch are very different (worse) from the ones from the original paper.

---

> ### Author Response · Authors · 2020-11-15
> **[Author's response] Thank you so much for the constructive feedback!**
>
> Dear reviewer,
>
> We would like to thank you so much for the constructive feedback. We respond to your concerns as follows:
> 1. For the concern related to the effectiveness of RankingMatch over FixMatch:
> - In comparison with FixMatch, our models can achieve competitive results even trained for **fewer epochs**. With the same batch size, FixMatch’s models were trained for **1024 epochs** to obtain the state-of-the-art results while our models were trained for only **128** and **256 epochs**, leading to a **faster training time**. For instance, we achieve the state-of-the-art results on CIFAR-10 with 250 labels (Table 2), CIFAR-100 with 10000 labels (Table 2), and SVHN with 250 and 1000 labels (Table 3).
> - Moreover, we also show the results when FixMatch is trained for only 128 epochs. As shown in Table 1, 3, and 4, RankingMatch obtains the better results compared to FixMatch over all cases in the case of trained for 128 epochs. Especially on CIFAR-10, both our two versions ($RankingMatch^{BM}$ and $RankingMatch^{CT}$) outperform FixMatch (Table 1). For instance, on CIFAR-10 with 40 labels, $RankingMatch^{BM}$ and $RankingMatch^{CT}$ improve the error rate by 4.20% and 2.76% compared to $Fixmatch^{RA}$, respectively (Table 1).
> 2. For the question related to the motivation of introducing auxiliary metric learning losses to be added to FixMatch:
> - Actually, FixMatch is a simple combination of existing SSL approaches such as consistency regularization and pseudo-labeling. FixMatch, as well as the consistency regularization approach, only considers the **different perturbations** of the **same input**. The model should produce unchanged with the different perturbations of the same input, but it is **not enough**. Our work is to fulfill this shortcoming. Our main motivation is that the **different inputs** of the **same class** (for example, two different *cat* images) should also have the similar model outputs. We show that by **simply integrating** metric learning losses into FixMatch, we can obtain promising results (as quantitatively shown in Table 1, 2, 3, and 4) and might open a new research direction for semi-supervised learning, where metric learning losses are deeply studied.
> - Moreover, we also visualize the “logits” space of FixMatch and RankingMatch, as shown in **Figure 2**. As you can see, by adding metric learning losses, the way, that the model clusters the samples, is changed, leading to **the different shapes of the clusters**. This would be interesting research for the community in the future.
> 3. For our statement: "We argue that the images from the same class do not have to have similar representations strictly, but their model outputs should be as similar as possible":
> - Our work aims to solve the image classification task. Basically, the model for image classification consists of two main parts: feature extractor and classification head. The feature extractor is responsible for understanding the image and generates the image representation. The image representation is then fed into the classification head to produce the model output (the scores for all classes).
> - If the feature extractor can generate the **very similar** image representations for the images from the same class, it will be good for the classification head. Otherwise, if these image representations are **not totally similar**, the classification head will have to pay more effort to produce the similar model outputs for the same-class images. Therefore, we can see that **the model outputs somehow depend on the image representations**.
> - As we can see for image classification, **the goal is to get the similar model outputs** for the same-class images **even when the image representations are not totally similar**. That is the main motivation for us to **apply metric learning losses directly to the model outputs**. To consolidate our statement, we do the visualization on the CIFAR-10 dataset, as shown in the link: https://drive.google.com/file/d/1Etj-snbbKdD1cSg81mk3kkcIutxnt13J/view?usp=sharing
> - The above figure shows the visualization of RankingMatch for the image representations and the model outputs. As illustrated in Figure b, the model outputs of the samples from the same class are clustered relatively well (*the same color means the same class*). As a result, the image representations of the same-class samples are **also clustered relatively well**, as shown in Figure a. Consequently, by forcing the model outputs of the same-class samples to be as similar as possible, we obtain the similar image representations as well.
>
> *To be continued in the following response due to the limited number of characters...*

---

> > ### Author Response · Authors · 2020-11-15
> > **[Author's response] Continued**
> >
> > To be concluded:
> > - We would like to remind you about one of our main contributions, BatchMean Triplet loss. Our proposed BatchMean Triplet outperforms two existing versions of Triplet loss (BatchAll and BatchHard), as shown in the ablation study (Table 5). In the ablation study, we also show the effectiveness of using L2-normalization in our method. It would also be found interesting via our visualization for BatchAll, BatchHard, and our BatchMean Triplet loss, as shown in Figure 8 in Appendix.
> > - We will add more explanations, including the visualization, for our statement to the paper (might be in Appendix due to the limited number of pages in the main text).
> > - We will update the results on another dataset like **Tiny-Imagenet**.
> > - We hope you can reevaluate our paper and reconsider your decision based on our efforts.
> >
> > Thank you so much for spending time reviewing our paper, and we are looking forward to hearing from you!

---

> ### Author Response · Authors · 2020-11-24
> **[Author's response] Paper and answers updated!**
>
> Dear reviewer,
>
> According to your concerns, our paper and answers are updated as follows:
> 1. We added the **detailed explanation** for our motivation and argument to **Appendix A** (**Details of Our Motivation and Argument**). We also added one sentence to the **fourth paragraph** of **Section 1** (**Introduction**) to consolidate our motivation: *“Consistency regularization approach incites the model to produce unchanged with the different perturbations of the same input, but this is not enough”*.
> 2. We added the results for the Tiny ImageNet dataset to **Section 4.4** (**Tiny ImageNet**) to prove that our method can work on a larger dataset.
>
> Thank you so much for reading, and we hope you can reconsider and reevaluate our paper!
>
> **Notably**, because our paper is updated, some figures have the number changed. Therefore, we updated the figure number in the previous response.

---

### Official Review · AnonReviewer2 · 2020-10-27
**Interesting idea but would require more in depth study**

**Rating:** 5
**Confidence:** 3

**Review:**

This paper proposes a novel method, called RankingMatch to the problem of semi-supervised learning (SSL). It extends known consistency regularization - based SSL methods by adding an additional loss that encourages similarity of outputs for samples of same class. In addition to, the paper proposes a new version of the triplet loss, called Batch Mean Triplet loss (BMT).

In general, there are some innovative and novel parts in this work, but in total it is somewhat incremental since big parts of the method and the results result from FixMatch, which serves as the basis for this method. The experimental results are promising, although not conclusive, for instance with larger amount of classes in cifar100, a different version of the method works better compared to a dataset with less classes, cifar10 (see Tables 1 and 2).

Pros:

The proposed idea to regularize the class output consistency between samples of the same class is interesting and should be studied further by the research community.

Another interesting idea is the BatchMean triplet loss that seems to be novel and could also be utilized in other settings.


Cons:

The clarity of the paper could be improved by clearly stating the differences and similarities to the closest method (I assume FixMatch in this case) in other parts than the intro as well. This applies to all variants of the proposed RankingMatch method.

One problem I see in this work is that the method is based on a relatively ad-hoc idea of class output similarity between samples of the same class. This assumption might hold for some datasets and not for others - this should have been studied in itself for various datasets. Definitely there may be correlation for many datasets, but one could imagine that for instance in tasks with big amount of classes, the correlation could sometimes be relatively low. For instance, in Imagenet, the output could include other objects that accidentally happen to be in the image together with the main object. Anyway it is interesting to see it works to some extent with the simple datasets, but it should be tested with Imagenet like was done with FixMatch.

Another problem is the batch-specific nature of the proposed loss variants that require also samples from the same class. This is not typically a problem in non-class-specific contrastive losses, but in this case, especially when the amount of classes increase (e.g., 1k or 10k), the proposed may start to perform worse when the same-class samples co-appear less often in the same batch.

Edit: I have read the changes and the author responsed, but have decided the keep the score.

---

> ### Author Response · Authors · 2020-11-15
> **[Author's response] Thank you so much for the constructive feedback!**
>
> Dear reviewer,
>
> We would like to thank you so much for the constructive feedback. To respond to your concerns, we answer the comments, which were mentioned in the second paragraph and the cons of the review, as follows:
> 1. The differences in performance between two versions of our method ($RankingMatch^{BM}$ and $RankingMatch^{CT}$) when the amount of classes is increased (for example, from CIFAR-10 to CIFAR-100):
> - We already mentioned this difference in Section 4.2 of our paper (Results with same settings). $RankingMatch^{BM}$ uses Euclidean distance as the similarity measurement, while $RankingMatch^{CT}$ uses cosine similarity. When the number of classes is increased, the dimension of the “logits” score grows up as well. Even though there is no perfect metric to measure the correlation of the vectors, we would like to say that the data will be more complex when the dimension is increased, so measuring the correlation between two different vectors in terms of the Euclidean distance may not be a suitable choice, so we decided to use another metric (cosine similarity in our case).
> - As a result, we empirically showed that in a high-dimensional space (for example, CIFAR-100), cosine similarity is more suitable than Euclidean distance in **most cases**. For instance, in Table 1 (CIFAR-10), Table 2 (CIFAR-10), Table 3 (SVHN), and Table 4 (STL-10), where the “logits” scores are the 10-dimensional vectors, $RankingMatch^{BM}$ is better than $RankingMatch^{CT}$ in all cases except SVHN with 1000 labels. On the other hand, in Table 1 (CIFAR-100) and Table 2 (CIFAR-100), where the “logits” scores are the 100-dimensional vectors, $RankingMatch^{CT}$ is better than $RankingMatch^{BM}$ in all cases except the case of 400 labels (the number of labeled data is relatively small). Realizing this difference and correlation, we think that if we can combine Euclidean distance and cosine similarity in an appropriate way, the performance might be improved, as mentioned in our conclusion section.
> 2. RankingMatch should be tested with ImageNet like was done in FixMatch:
> - Unfortunately, as mentioned in the paper, our computing resources are limited, so it is very hard for us to conduct experiments on a huge dataset like ImageNet. However, to respond to your suggestion, we **are implementing** and **doing experiments with a smaller version of ImageNet**, called **Tiny-ImageNet**. We will **report the results in the later response** and **update them in our paper**.
> 3. The problem of batch-specific nature where the same-class samples co-appear less often in the same batch:
> - We strongly agree with this point, and we actually thought about this kind of problem when designing our method. However, in the case of $RankingMatch^{CT}$ using contrastive loss, our model is trained to **maximize the similarity of positive samples** (samples from the same class) within a batch, and the number of positive samples in the batch does not affect performance significantly. For example, even though there are only **two positive samples** in a batch, $RankingMatch^{CT}$ still aims to **maximize the similarity between them**. When the number of classes is increased (from CIFAR-10 to CIFAR-100), the performance of RankingMatch highly depends on the similarity measurement (as mentioned above, in our first answer). For instance, $RankingMatch^{CT}$ becomes more efficient than $RankingMatch^{BM}$ on the CIFAR-100 dataset except in the case of 400 labels (where the number of labels is relatively small).
>
> To be concluded:
> - We will add the description of the differences and similarities between RankingMatch (including $RankingMatch^{BM}$ and $RankingMatch^{CT}$) and FixMatch in another part (might be Appendix due to the limited number of pages in the main text).
> - We will update the results on the Tiny-ImageNet dataset once it is completed.
> - We hope you can reevaluate our paper based on our efforts.
>
> Thank you so much for spending time reviewing our paper, and we are looking forward to hearing from you!

---

> ### Author Response · Authors · 2020-11-24
> **[Author's response] Paper and answers updated!**
>
> Dear reviewer,
>
> According to your concerns, our paper and answers are updated as follows:
> 1. We added the detail of the comparison between our method and the others (including the closest method, FixMatch) to **Appendix B** (**Comparison of Methods**) in our paper.
> 2. We completed doing experiments on the Tiny ImageNet dataset and updated the results to **Section 4.4** (**Tiny ImageNet**) in our paper. The detail of Tiny ImageNet, including labeled/unlabeled data division, was also added to Appendix D.4 (Dataset Details).
>
> Thank you so much for reading, and we hope you can reconsider and reevaluate our paper!

---

### Official Review · AnonReviewer1 · 2020-10-28
**Review of RankingMatch**

**Rating:** 4
**Confidence:** 5

**Review:**

This paper considers the problem of semi-supervised learning, where the training set contains both labeled images and unlabeled images. To address this task, the paper proposes a method called RankingMatch which exploits the idea that input images having the same label should have similar model outputs. To incorporate this idea into training, the paper develops a BatchMean Triplet loss to guide the model learning from the unlabeled images. Experiments are conducted on the CIFAR-10, CIFAR-100, SVHN, and STL-10 datasets. Results of RankingMatch show some improvements over competing methods in some cases.

Quality: The presentation of this paper is clear. Figure 1 clearly illustrates the pipeline of the framework. This paper reads well and is easy to follow.

Clarify: This paper clearly presents the proposed method and the motivation of each design choice is clearly described. While some of the components are from existing papers, the paper clearly acknowledges the source. This allows me to understand which parts of the method are built upon existing approaches and helps me better identify the contributions and new components of this paper.

Originality: Generally, the idea of this paper makes sense. However, there are several parts similar (if not the same) to existing methods. The cross-entropy loss in Equation 2 is a standard loss function that is commonly adopted in classification tasks. The loss in Equation 3 is from FixMatch (mentioned in the paper as well). The triplet loss and contrastive loss are two loss functions that are widely applied in recognition and representation learning tasks. All of these loss functions to me are standard and common knowledge in the community. The idea of applying ranking loss on the model outputs is highly similar to a recent paper published in ECCV 2020 [a]. The idea of learning from the unlabeled data in [a] is to leverage class-wise similarity scores to form a representation and use this representation to describe an unlabeled data. For a pair of unlabeled images, [a] uses the class-wise similarity representation to assign positive/negative labels to that pair of images by comparing the distance between the two similarity representations. After assigning pseudo labels to that pair of images, [a] applies contrastive loss to further learn from the unlabeled data. The idea in this paper is very similar. The output of the model (classifier) can also be viewed as a form of class-wise similarity distributions. Namely, for each unlabeled image, the classifier outputs a distribution that indicates the similarity of that input image to each class of the labeled set. To me, the high-level idea is similar. However, the authors did not acknowledge the similarity with [a] in the submission. This will make readers feel like the idea of applying ranking loss on the model outputs is original in this paper.

[a] Chen et al. Learning to learn in a semi-supervised fashion. In ECCV, 2020. https://arxiv.org/pdf/2008.11203.pdf

Significance: Given that some parts of this paper are highly similar to [a], the significance of this paper is downplayed. Also, there are several components in this paper are from existing/well-known papers. Based on these grounds, the contribution of this paper is limited.

Request for author response: I would like to see how the authors compare their paper with [a] in terms of the idea of class-wise similarity and representation learning. In particular, I would like to know the pros and cons of this paper compared to [a]. It would be great if the authors can talk about whether the semantics-oriented similarity representation in [a] could be used (and how to use it) to help improve the performance of the proposed method in the setting concerned by the paper.

Rating: Given that there are many parts similar to [a] and they are not acknowledged in the paper and many of the components in the paper are from existing approaches, I can only rate 4 for this submission at this point. I will reevaluate this paper after seeing the reviews from the other reviews as well as the author response.

---

> ### Author Response · Authors · 2020-11-13
> **[Author's response] Thank you so much for the constructive feedback!**
>
> Dear reviewer,
>
> First of all, we would like to thank you so much for your constructive feedback and the suggested paper [a]. To be honest, it was our shortcoming when omitting [a]. If the model output (in our case) is viewed as a form of class-wise similarity distributions where each its component indicates the similarity to the corresponding class, we agree with the argument that the high-level idea of our method is similar to [a], but it should be very high level. To respond to your comments, we answer three requested questions as follows:
> 1. **Comparison of our paper with [a] in terms of the idea of class-wise similarity and representation learning**
> - In [a], the class-wise similarity representations are manually built upon the model outputs (feature representations in this case) with the help of the similarity measurement (cosine similarity or L2 distance).
> In our method, we implicitly obtain class-wise similarity thanks to the classification head, and the model itself is responsible for generating class-wise similarity.
> - Due to the difference of the specific tasks, we use class-wise similarity scores (model outputs) to produce the semantic pseudo-labels (for example, *dog* or *cat*) while class-wise similarity scores in [a] are to assign pseudo positive/negative labels
> 2. Actually, it is not easy to directly show the pros and cons of our method compared to [a] because the specific task is different (our work aims to solve image classification while [a] deals with image retrieval and person re-identification). However, **in terms of the idea of class-wise similarity representations, we provide pros and cons as follows**.
> *Pros:*
> - If the model output (in our case) is viewed as a form of class-wise similarity distributions, the class-wise similarity representations in our case consider **all classes**. In contrast, the class-wise similarity representations in [a] only contain **sampled classes** at **a time**. However, as our understanding, [a] tried to cover all classes by conducting multiple episodes.
> - Because [a] computed the class-wise similarity representations based on the selected reference images of the sampled classes, the performance might **somewhat depend on this selection**. For instance, different reference images lead to different class-wise similarity representations. Our work does not have this kind of problem. We train the model to obtain the class-wise scores for all classes.
> - Our model is trained in a **single training process** with both labeled and unlabeled data, while [a] consists of **two phases** (Section 3 in [a]). In the first phase, [a] only uses labeled data. In the second phase, [a] utilizes the learned concept in the first phase to train the model with both labeled and unlabeled data.
> *Cons:*
> - We did not consider the case where labeled and unlabeled data share **non-overlapping categories**. However, this might be excusable because our method aims to solve the image classification task, so the classes should be determined and provided by labeled data set.
> 3. **How to apply the semantics-oriented similarity representation in [a] to our method to improve performance**.
> For this question, we made a **PDF file** at
>  https://drive.google.com/file/d/19L6cyNZ9oFWzh2aXQ0Y7VrHIWX4AEumB/view?usp=sharing
> to explain our idea to integrate the semantics-oriented similarity representation into our method.
>
> Additionally, we would like to remind you about our contributions. In addition to the combination of semi-supervised learning and metric learning, one of our main contributions is proposing a new version of Triplet loss, called **BatchMean Triplet loss**. Our proposed version outperforms BatchHard and BatchAll, as shown quantitatively in the ablation study. Moreover, we also provide the **visualization** in the **Appendix**, and we hope it might help consolidate our contribution.
>
> To be concluded:
> - [a] is very interesting and good work, and we will include [a] to related work in our paper.
> - We hope you can reconsider your decision. This work is our attempt to unify the idea of consistency regularization semi-supervised learning and metric learning. Moreover, our BatchMean Triplet loss might be effectively applied to other tasks (image retrieval, person re-identification) to improve the performance.
>
> Thank you so much for spending time reviewing our paper, and we are looking forward to hearing from you!
>
> [a] Yun-Chun Chen, Chao-Te Chou, and Yu-Chiang Frank Wang. Learning to learn in a semi-supervised fashion. In ECCV, 2020. https://arxiv.org/pdf/2008.11203.pdf

---

> ### Author Response · Authors · 2020-11-24
> **[Author's response] Paper and answers updated!**
>
> Dear reviewer,
>
> According to your concerns, our paper and answers are updated as follows:
> 1. We included [a] in **Section 2** (**Related Work - Metric Learning and Ranking Loss**) and presented the main similarity and differences between our paper and [a]. More comparisons between our method and [a] were presented in the previous response.
> 2. For how to apply the semantics-oriented similarity representation in [a] to our method, we completed running experiments and updated the results at
> https://drive.google.com/file/d/16tyKop7uy4HL9kHCSF8S_gTLLfGlZKWD/view?usp=sharing
>
> Thank you so much for reading, and we hope you can reconsider and reevaluate our paper!
>
> [a] Yun-Chun Chen, Chao-Te Chou, and Yu-Chiang Frank Wang. Learning to learn in a semi-supervised fashion. In ECCV, 2020. https://arxiv.org/pdf/2008.11203.pdf

---

### Author Response · Authors · 2020-11-24
**[Author's general response] Paper updated!**

Dear,

We would like to thank reviewers for the constructive comments and feedback. We also would like to give thanks to Area Chairs and Program Chairs for their effort. Although we made the response for each reviewer, we summarize main updates to our paper as follows:
1. **Section 1** (**Introduction**). We added one sentence to the fourth paragraph to consolidate our motivation: *“Consistency regularization approach incites the model to produce unchanged with the different perturbations of the same input, but this is not enough”*.
2. **Section 2** (**Related Work - Metric Learning and Ranking Loss**). We added and presented the main similarity and differences between our paper and the related work [a].
3. **Section 4.4** (**Tiny ImageNet**). We added the experimental results to verify the performance of our method on a larger dataset, Tiny ImageNet.
4. **Section 4.6** (**Qualitative results**). We added the visualization of our method and other methods for comparison.
5. **Section 4.6** (**Computational efficiency**). We added the quantitative measurement to show how much our BatchMean Triplet loss is computationally efficient.
6. **Appendix A** (**Details of Our Motivation and Argument**). We added the detailed explanation for our motivation and argument presented in Section 1.
7. **Appendix B** (**Comparison of Methods**). We added the detailed comparison between our method and other methods.
8. **Appendix F** (**Computational Efficiency of BatchMean Triplet loss**). We added more quantitative results to show how much our BatchMean Triplet loss is computationally efficient.

Thank you so much for reading and spending time reviewing our paper, and we hope you can reconsider and reevaluate our paper!

[a] Yun-Chun Chen, Chao-Te Chou, and Yu-Chiang Frank Wang. Learning to learn in a semi-supervised fashion. In ECCV, 2020. https://arxiv.org/pdf/2008.11203.pdf

---

### Decision · Program_Chairs · 2021-01-07
**Final Decision**

**Decision:**

Reject

**Comment:**

Despite the performance gains of RankingMatch over the benchmarks used in the paper, the reviewers remained concerned about how the paper compares to state of the art in several respects.